# ORP2 couples LDL-cholesterol transport to FAK activation by endosomal cholesterol/PI(4,5)P₂ exchange

Kohta Takahashi[1,2,‡], Kristiina Kanerva[1,2,†] (iD), Lauri Vanharanta[1,2,†], Leonardo Almeida-Souza[3,4,5], Daniel Lietha[6] (iD), Vesa M Olkkonen[2] & Elina Ikonen[1,2,*] (iD)

## Abstract

Low-density lipoprotein (LDL)-cholesterol delivery from late endosomes to the plasma membrane regulates focal adhesion dynamics and cell migration, but the mechanisms controlling it are poorly characterized. Here, we employed auxin-inducible rapid degradation of oxysterol-binding protein-related protein 2 (ORP2/OSBPL2) to show that endogenous ORP2 mediates the transfer of LDL-derived cholesterol from late endosomes to focal adhesion kinase (FAK)-/integrin-positive recycling endosomes in human cells. *In vitro*, cholesterol enhances membrane association of FAK to PI(4,5)P₂-containing lipid bilayers. In cells, ORP2 stimulates FAK activation and PI(4,5)P₂ generation in endomembranes, enhancing cell adhesion. Moreover, ORP2 increases PI(4,5)P₂ in NPC1-containing late endosomes in a FAK-dependent manner, controlling their tubulovesicular trafficking. Together, these results provide evidence that ORP2 controls FAK activation and LDL-cholesterol plasma membrane delivery by promoting bidirectional cholesterol/PI(4,5)P₂ exchange between late and recycling endosomes.

**Keywords** cholesterol trafficking; focal adhesion kinase; oxysterol-binding protein-related protein; phosphoinositides; recycling
**Subject Categories** Membranes & Trafficking; Metabolism
**The EMBO Journal (2021) 40: e106871**

## Introduction

Mammalian cells acquire cholesterol, a vital component of cell membranes, via receptor-mediated uptake of low-density lipoprotein (LDL) (Brown & Goldstein, 1986; Ikonen, 2008). The LDL-derived cholesterol is released in the acidic endosomal milieu and two late endosomal/lysosomal proteins, Niemann–Pick C1 (NPC1) and Niemann–Pick C2 (NPC2), act in tandem to transfer cholesterol from the organelle lumen to the limiting membrane (Kwon *et al*, 2009; Li *et al*, 2016; Qian *et al*, 2020). The LDL-derived cholesterol is then transported to the plasma membrane (PM), to serve a structural role as a major PM constituent. PM cholesterol can be divided into "accessible" and "inaccessible" pools based on its accessibility to cholesterol-binding toxins (Das *et al*, 2014; Infante & Radhakrishnan, 2017). Cholesterol hydrolyzed from LDL in endo-lysosomes adds to the accessible PM cholesterol pool (Das *et al*, 2014).

How LDL-cholesterol reaches the PM is not well understood. We have reported that NPC1 organelles and LDL-cholesterol are transported toward the cell periphery via Rab8a-MyosinVb-actin-dependent membrane trafficking (Kanerva *et al*, 2013). This route delivers LDL-cholesterol to the PM in the proximity of focal adhesions (FAs), stimulating FA dynamics and cell migration (Kanerva *et al*, 2013). Which proteins are involved as actual cholesterol transporters along this route after NPC1, is not clear. Moreover, the precise interactions by which LDL-cholesterol facilitates cell migration remain largely unknown. The available evidence indicates that cholesterol not only modulates membrane order in integrin-mediated adhesion sites (Gaus *et al*, 2006; Lietha & Izard, 2020) but also affects integrin recycling pathways (Reverter *et al*, 2014).

Several proteins with cytoplasmic cholesterol-binding domains have been implicated in late endosomal cholesterol trafficking, but none specifically at the stage of post-NPC1 LDL-cholesterol transport leading to the PM. ORP2/OSBPL2 is a candidate for this, as it has been related to cholesterol transport involving the PM (Laitinen *et al*, 2002; Hynynen *et al*, 2005; Jansen *et al*, 2011; Koponen *et al*, 2019; Wang *et al*, 2019). Moreover, ORP2 can exchange cholesterol for PI(4,5)P₂, a PM enriched phosphoinositide (Wang *et al*, 2019). According to recent data, PM cholesterol content was decreased and

1   Department of Anatomy and Stem Cells and Metabolism Research Program, Faculty of Medicine, University of Helsinki, Helsinki, Finland
2   Minerva Foundation Institute for Medical Research, Helsinki, Finland
3   Helsinki Institute of Life Science, HiLIFE, University of Helsinki, Helsinki, Finland
4   Faculty of Biological and Environmental Sciences, University of Helsinki, Helsinki, Finland
5   Institute of Biotechnology, University of Helsinki, Helsinki, Finland
6   Centro de Investigaciones Biológicas Margarita Salas (CIB), Spanish National Research Council (CSIC), Madrid, Spain
    *Corresponding author. Tel: +358 50 448 5050; E-mail: elina.ikonen@helsinki.fi
    †These authors contributed equally to this work.
    ‡Present address: Laboratory of Microbiology and Immunology, Graduate School of Pharmaceutical Sciences, Chiba University, Chiba, Japan

$PI(4,5)P_2$ increased in ORP2 knockout HEK293 cells (Wang *et al*, 2019), whereas in ORP2-silenced endothelial cells, PM $PI(4,5)P_2$ was increased without a significant change in the cholesterol amount (Koponen *et al*, 2020). This leaves open the precise lipid transport step(s) mediated by ORP2 and whether the primary function of ORP2 would be to control the level of cholesterol or $PI(4,5)P_2$ in the PM.

The challenges in assigning specific physiological lipid transport steps for ORP2 in cells relate to the facts that endogenous ORP2 is largely cytoplasmic (Laitinen *et al*, 2002) and that the lipid transfer function of ORP2 does not require membrane anchoring (Wang *et al*, 2019). At the organismal level, human patients carrying ORP2 mutations have been reported (Thoenes *et al*, 2015; Xing *et al*, 2015; Wu *et al*, 2019), but they present with an unexpected phenotype, autosomal dominant hearing loss, without providing an obvious indication to the physiological function of the protein. In an attempt to pinpoint the cellular lipid transfer steps that endogenous ORP2 is responsible for, we have employed CRISPR-based gene editing to tag endogenous ORP2 for fluorescent imaging and for acute protein degradation, using a system we have recently developed (Li *et al*, 2019) and applied for studying subcellular lipid trafficking (Salo *et al*, 2019).

Our studies provide evidence that LDL-derived cholesterol spreads rapidly from late to recycling endosomal compartments on its way to the PM. The transfer of cholesterol from late to recycling endosomes is facilitated by ORP2 which counterexchanges it for $PI(4,5)P_2$. The ORP2-mediated bidirectional lipid exchange serves for important cross-talk between late and recycling endosomes: LDL-derived accessible cholesterol is transferred to the PM not only via NPC1 late endosomes but also via FAK/integrin recycling endosomes where it increases FAK activity. FAK activity and the generated $PI(4,5)P_2$ are in turn needed for maintaining NPC1 organelle dynamics.

## Results

### LDL enhances FA dynamics in a FAK-dependent manner

To visualize accessible cholesterol in cells upon LDL loading, we purified a fluorescently tagged cholesterol-binding domain (D4H)

derived from Perfringolysin O (Maekawa & Fairn, 2015; Wilhelm *et al*, 2017). The purified D4H bound specifically to cholesterol-containing membranes (Fig EV1A). We then labeled A431 cells on ice with the fluorescently tagged recombinant protein, fixed, and imaged, essentially as previously described (Wilhelm *et al*, 2017) (Fig 1A). In LDL-depleted conditions, D4H binding to the PM was significantly decreased compared with complete medium, and upon increasing times of LDL loading the signal gradually increased (Fig 1B and C), in line with the notion that LDL-derived cholesterol adds to the accessible cholesterol pool (Das *et al*, 2014).

To examine the role of LDL-cholesterol in FA dynamics, we generated cell lines stably expressing the FA proteins GFP-paxillin or mCherry-talin and quantified FA assembly and disassembly rates upon LDL loading (Figs 1D and E, and EV1B). We found that in both cell lines LDL enhanced the assembly and disassembly of FAs in a FAK-dependent manner, as the FAK inhibitor PF-228 or overexpression of a kinase dominant-negative mutant FAK Y397F, suppressed the LDL-induced FA dynamics (Figs 1E and EV1C). Instead, cholesterol removal by methyl-β–cyclodextrin (MβCD) inhibited FA assembly and disassembly (Fig EV1D). The LDL-induced stimulation of FA dynamics was observed from 1 h of LDL loading onwards and it was abolished by NPC1 inhibition (Fig EV1E).

We also found that addition of LDL to LDL-depleted cells induced an increase in FAK phosphorylation starting from ~ 0.5 h (Fig 1F). In comparison, addition of cholesterol to cells directly from cholesterol/cyclodextrin complex, which increases membrane cholesterol content within minute(s), induced an increase in FAK phosphorylation within minutes (Fig EV1F). Together, these results are compatible with the idea that cholesterol enhances FAK activation and that in cells, LDL-derived cholesterol might reach FAK before its levels are prominently increased at the cell surface.

### LDL-cholesterol is delivered to FAK/integrin-containing endosomes

In migrating cells, FAK co-localizes with recycling integrins, sustaining them in the active conformation and polarizing them toward the leading edge (Nader *et al*, 2016). We therefore studied how the organelles containing active FAK localize relative to NPC1-harboring

---

**Figure 1.  LDL enhances FA dynamics in a FAK-dependent manner.**

A   Schematic outline of plasma membrane D4H labeling. CM and LPDS indicate complete medium (10% FBS in DMEM) and 5% LPDS in DMEM, respectively.

B   After 24-h LDL depletion in 5% LPDS, cells were loaded with 50 μg/ml LDL for the indicated times. Plasma membrane cholesterol was labeled with D4H and imaged with widefield epifluorescence microscopy.

C   Analysis of (B). Mean ± SD, *n* = 15 fields of cells as in (B) pooled from 2 independent experiments.

D   Representative FA lifetime in cells treated without or with 50 μg/ml LDL (−LDL or +LDL) for 1–2 h after 24 h in 5% LPDS. Time-lapse imaging of GFP-paxillin dynamics in cells was acquired by TIRF microscopy and maximum intensity during FA lifetime was normalized to 1.

E   Analysis of FA assembly and disassembly rate of cells incubated +/−LDL as in (D). Where indicated, 10 μM PF-228 was added after 30 min of LDL loading. Assembly rate: *n* = 290 FAs (−LDL); 410 (+LDL); 192 (+LDL+PF-228) and disassembly rate: *n* = 397 (−LDL); 502 (+LDL); 214 (+LDL+PF-228) pooled from 4 independent experiments. The Tukey box plots represent the median and 25th and 75th percentiles (interquartile range). The whiskers show the highest and lowest observations. Non-parametric Kruskal–Wallis analysis of variance with Bonferroni error correction.

F   Representative immunoblot and quantification of pFAK levels in 5% LPDS (−LDL) and upon increasing LDL loading times. Mean ± SD, *n* = 3, paired Student's *t*-test (*$P < 0.05$).

G   Representative confocal images of NPC1, internalized integrin β1, and pFAK localization. Stably NPC1-GFP-expressing cells were loaded with 2 μg/mL integrin β1 antibody for 30 min before fixation and stained with pFAK antibody followed by secondary antibodies. Dashed line indicates the cell edge.

H   Quantification of (G) using Mander's colocalization for the indicated markers. Mean ± SD, *n* = 5 cells. Cells were treated with 5% LPDS for 1 day and then without (−) or with (+) 50 μg/ml LDL for 1h.

Source data are available online for this figure.

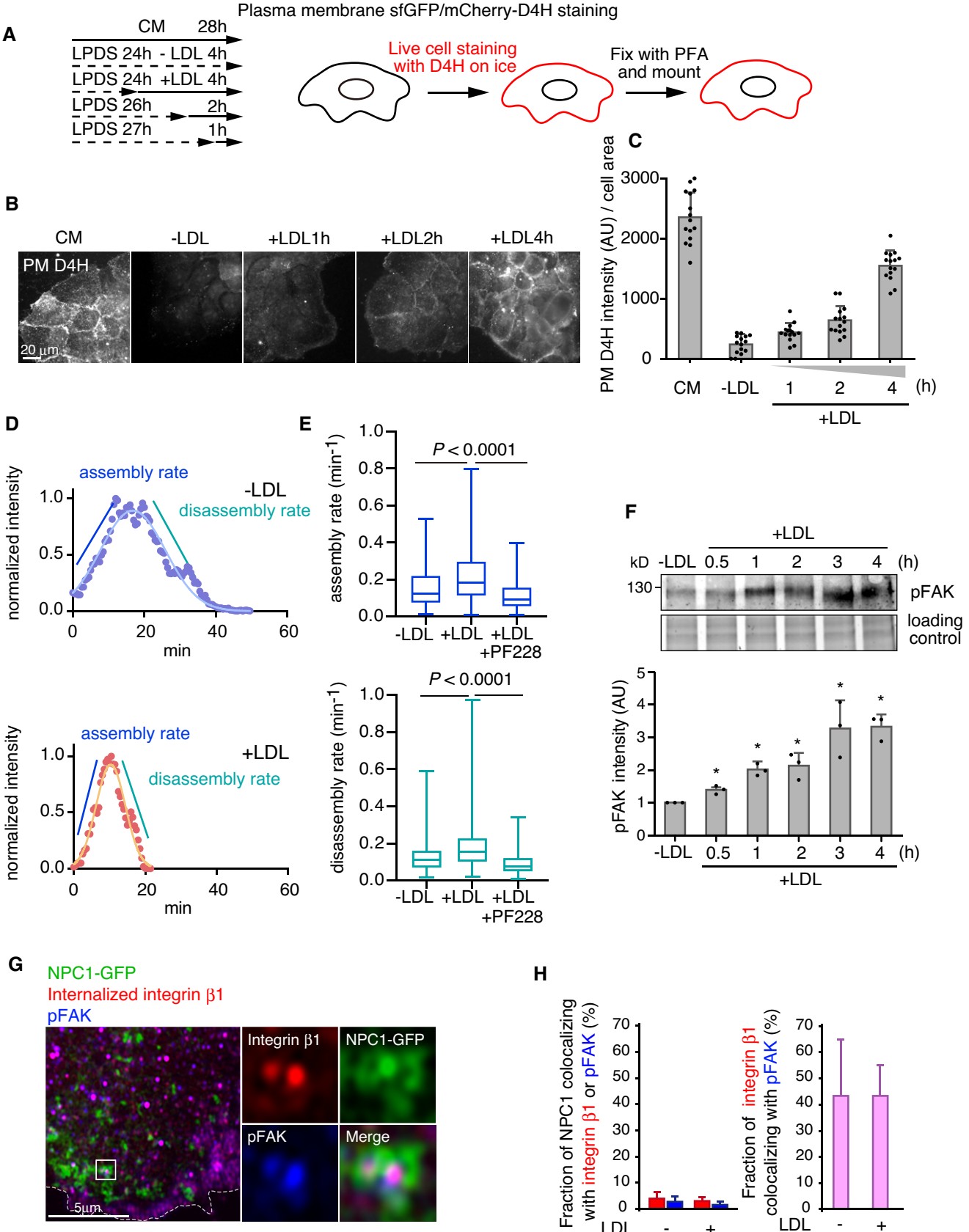

**Figure 1.**

organelles upon LDL loading. Cells stably overexpressing NPC1-GFP were LDL-depleted and then loaded with LDL for 1–2 h, of which the last 30 min in the presence of antibodies against integrin β1, to trace the recycling integrins. Cells were then immunostained with pFAK antibodies (Appendix Fig S1). pFAK and internalized integrin β1 were often in overlapping vesicular structures as expected, but NPC1 organelles did not colocalize with them (Fig 1G and H). Yet, the NPC1 and FAK/integrin-containing endosomes were often in close proximity to each other, with NPC1 endosomes sometimes encircling pFAK/integrin-containing organelles (Fig 1G).

We have earlier reported that LDL particles labeled with BODIPY-cholesteryl linoleate can be used to follow the export of the fluorescent sterol from late endosomes and its delivery to the PM or lipid droplets (Kanerva *et al*, 2013; Heybrock *et al*, 2019). Using a similar strategy, we explored whether the LDL-derived fluorescent sterol can be delivered to integrin β1-containing recycling endosomes visualized by internalized fluorescently labeled antibodies (Fig 2A). After 2 h labeling of cells with BODIPY-cholesteryl linoleate/LDL, most of the label was in late endosomal compartments labeled with fluorescent dextran (Fig EV2A), as previously shown (Kanerva *et al*, 2013). However, roughly 20% of LDL-derived BODIPY-cholesterol colocalized with recycling integrin β1 and this fraction increased to ~ 60% during 1 h of chase (Figs 2B and EV2B). Considering that integrin β1 may also be found in late endosomes, at least when fibronectin is internalized (Lobert *et al*, 2010), we also assessed the localization of LDL-derived BODIPY-cholesterol relative to another recycling protein, transferrin. Similarly, increasing colocalization between BODIPY-cholesterol and fluorescent transferrin was observed at early times of LDL chase (Fig EV2C–E). These data suggest that the fluorescent sterol deriving from LDL rapidly reaches recycling endosomal compartments.

Fast frame-rate live-cell imaging revealed rapid and repeated kiss-and-run type of contacts between the BODIPY-cholesterol and integrin β1-containing organelles (Fig EV2F and Movie EV1). These contacts were apparently engaged in sterol cargo exchange, as integrin β1-positive organelles showed increased BODIPY-cholesterol fluorescence after contacting BODIPY-cholesterol enriched organelles (Fig EV2G).

To circumvent caveats related to the use of fluorescent sterol analogs, we also visualized intracellular, LDL-derived accessible cholesterol using the D4H probe, as previously described (Wilhelm *et al*, 2017) (Fig 2C). In LDL-depleted conditions, virtually no intracellular accessible cholesterol was detectable (Fig 2D and E). Upon increasing LDL loading (1–4 h), the intracellular D4H labeling increased, first appearing as punctate structures at the perinuclear region and then increasingly spreading toward the cell periphery (Fig 2D and E). With increasing LDL loading, the total D4H area overlapping with NPC1-GFP and especially with internalized integrin β1 increased (Fig 2F and G), while the fraction of D4H co-localizing with NPC1 decreased (Fig 2F and H), indicative of the delivery of accessible LDL-derived cholesterol to post-NPC1 destinations. These results agree with those obtained by following the itinerary of BODIPY-cholesteryl linoleate-labeled LDL and suggest that LDL-derived accessible cholesterol spreads from NPC1 endosomes toward the cell periphery reaching FAK/integrin endosomes on its way to the PM.

## FAK regulates cholesterol export from NPC1 endosomes and endomembrane PI(4,5)P₂ levels

We next investigated whether FAK activity is needed for LDL-cholesterol PM delivery, taking advantage of the FAK inhibitor PF-228, which blocks the cycle of FA turnover but did not inhibit LDL uptake (Fig EV3A). Loading of LDL-deprived cells with LDL in the presence of PF-228 resulted in reduced PM delivery of LDL-cholesterol as assessed by PM D4H labeling (Fig EV3B and C). Moreover, a prominent redistribution of the intracellular accessible LDL-derived cholesterol pool was observed by the intracellular D4H labeling protocol. Although the total D4H-labeled area increased upon LDL loading in PF-228-treated cells, the increase was smaller than in control cells (Fig 3A and B). The fraction of D4H overlapping with NPC1 failed to decrease efficiently over time; rather, there was a marked colocalization of NPC1-GFP and intracellular D4H even after 4 h of LDL loading, which was hardly detectable in the absence of the inhibitor (Fig 3A and C). In parallel, the spreading of D4H labeling to the integrin β1 compartment was attenuated (Fig 3A and D). Reduced PM delivery and late endosomal retention of LDL-cholesterol was also observed in cells where FAK was silenced (Figs 3E–G and EV3D and E), arguing that these phenotypes in PF-228 treated cells are not due to unspecific drug effects. Together, these findings provide evidence that LDL-derived accessible cholesterol fails to efficiently exit NPC1 organelles if FAK is not active.

While a role of FAK in endosomal cholesterol delivery was unexpected, FAK is known to control endomembrane PI(4,5)P₂

---

**Figure 2. LDL-cholesterol is rapidly delivered to FAK/integrin endosomes.**

A  Outline of live-cell BODIPY-cholesteryl linoleate/LDL labeling and chase combined with AF680-integrin β1 antibody labeling. Cells were pre-treated for 1 day in 5% LPDS.

B  Quantification of the fraction of BODIPY-cholesterol (BC) fluorescence in integrin β1-positive organelles at the indicated chase times. Exemplary images are shown in Fig EV2B. Mean ± SEM, *n* = 8 (0 min); 6 (15 min); 5 (30 min); 6 (45 min); and 7 (60 min) cells from 3 independent experiments.

C  Schematic outline of intracellular D4H labeling.

D  After 1-day LDL depletion in 5% LPDS, cells were loaded with 50 μg/ml LDL for the indicated times and intracellular cholesterol was detected with D4H as in (C) by confocal microscopy. CM, complete medium. Dashed lines indicate cell outlines.

E  Quantification of (D). Mean ± SD, *n* = 25–26 cells pooled from 2 independent experiments.

F  Cells were treated for 1 day in 5% LPDS followed by 50 μg/ml LDL loading for the indicated times. Representative confocal images of cells triple labeled with intracellular accessible cholesterol (D4H), NPC1-GFP and internalized integrin β1. Dashed lines indicate cell outlines.

G  Quantification of D4H area positive for NPC1 or internalized integrin β1 in a peripheral region toward the leading edge (exemplary box shown as inset in (F)).

H  Fraction of D4H area overlapping with NPC1 or internalized integrin β1 in a peripheral region toward the leading edge (exemplary box shown as inset in (F)).
   Mean ± SD, *n* = 27–30 cells pooled from 2 independent experiments. Student's *t*-test.

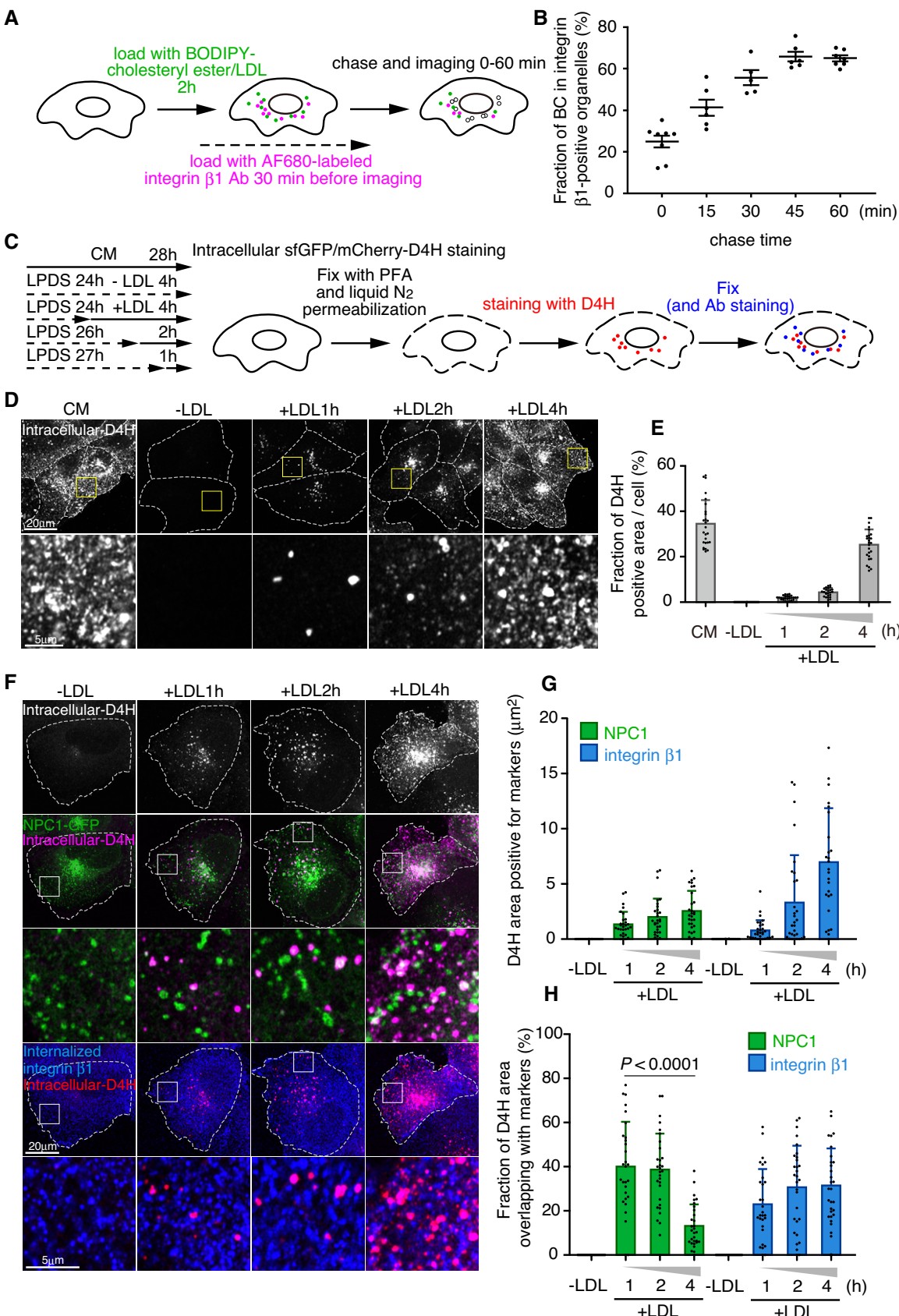

Figure 2.

generation by activating type I phosphatidylinositol phosphate kinase (PIPKIγ) on endosomes (Nader *et al*, 2016) (Fig 3H). We therefore investigated the effect of FAK on endomembrane PI(4,5)P$_2$ distribution using anti-PI(4,5)P$_2$ antibody staining, with a previously validated protocol (Hammond *et al*, 2009). This revealed that FAK inhibition with PF-228 indeed resulted in a pronounced reduction in endomembrane PI(4,5)P$_2$ levels (Fig 3I and J). Moreover, reduced PM PI(4,5)P$_2$ labeling upon PF-228 treatment was observed using the PLCdelta PH domain as a sensor for this phosphoinositide (Fig EV3F and G).

### Endogenous ORP2 localization and acute removal by auxin-inducible degradation

ORP2 has been implicated in cholesterol transport to the PM (Laitinen *et al*, 2002; Hynynen *et al*, 2005; Koponen *et al*, 2019; Wang *et al*, 2019). Accordingly, we found that ORP2 silencing by siRNA decreased the PM accessible LDL-derived cholesterol (Fig EV4A and B). Yet, these data do not pinpoint where the ORP2-mediated lipid exchange takes place. To investigate the site of action of endogenous ORP2, we tagged ORP2 at the genomic locus with GFP and degron cassettes (Appendix Fig S2), so that the endogenous GFP-tagged ORP2 can be visualized and acutely depleted using the auxin (indole-3-acetic acid, IAA)-inducible degron (AID) system (Nishimura *et al*, 2009; Li *et al*, 2019) (Fig 4A). With this system, IAA induces rapid ORP2 removal in cells harboring the complete AID machinery (ORP2-degron cells, > 95% reduction in endogenous ORP2 protein in ~ 30 min), while IAA had no effect in control cells lacking AtAFB2 expression (Fig EV4C).

When ORP2-degron and control cells were co-plated and then treated with IAA for 1 h, the specific signal from endogenous GFP-ORP2 in control cells was dim, but readily visible over the background (ORP2-depleted cells). GFP-ORP2 was observed as diffuse cytoplasmic fluorescence and as small mobile punctate structures, with no clear concentration in the PM (Fig 4B) or obvious change upon LDL depletion/loading of cells (not shown). To examine the relationship of endogenous GFP-ORP2 with NPC1 and FAK/integrin endosomes, we reduced the cytosolic GFP-ORP2 signal by saponin pre-permeabilization prior to fixation (Pfisterer *et al*, 2017), followed by anti-GFP (to enhance the remaining ORP2 signal) and pFAK, integrin β1, or NPC1 antibody labelings (Fig 4C–E). This revealed that about half of the endogenous ORP2 signal was in contact or overlapped with pFAK or integrin β1 and about 20% with

NPC1, suggesting that at steady state, a higher fraction of ORP2 was associated with integrin/FAK than NPC1-containing endosomes (Fig 4D). We also observed occasional concentrations of GFP-ORP2 between NPC1 and integrin β1-containing organelles, especially in the cell periphery (Fig 4E).

### Acute ORP2 removal impairs LDL-cholesterol delivery to the PM and cell adhesion

We next studied the effects of acute ORP2 depletion in this system. While LDL uptake was not affected by acute ORP2 removal (Fig EV4D), a clear reduction in PM accessible LDL-derived cholesterol was observed when IAA was added together with LDL, i.e., ORP2 was depleted in parallel with LDL loading for 1–4 h (Fig 4F and G). Instead, under these conditions no effect was observed in the delivery of LDL-cholesterol to the ER for suppression of SREBP2 target gene expression or for cholesterol re-esterification (Fig 4H–J). Considering that ORP2 exchanges PI(4,5)P$_2$, we also investigated whether acute ORP2 degradation alters PM PI(4,5)P$_2$. Based on PLCdelta PH-domain distribution, no obvious changes in PM PI(4,5)P$_2$ were observed during the first 4 h upon ORP2 removal; however, after 3 days of ORP2 ablation, more PM PI(4,5)P$_2$ was detected (Fig EV4E–G), in consistence with cells silenced for ORP2 for 3 days (Koponen *et al*, 2020). In addition, we tested whether a well-known PM PI(4,5)P$_2$-dependent membrane transport process, clathrin-mediated endocytosis of transferrin was affected, and found it not to be altered in 4 h of ORP2 removal (Fig EV4H). Together, these results suggest that an increase in PM PI(4,5)P$_2$ is not acute but develops gradually upon ORP2 removal.

We also assessed whether ORP2 removal affects integrin-dependent cell adhesion. We found that acute ORP2 depletion abolished the LDL-induced increase in cell adhesion (as measured by cell area at 1 h after replating, Fig EV4I–K). Of note, acute ORP2 removal also inhibited cell adhesion in the absence of LDL, indicating that the effect of ORP2 was not strictly LDL-dependent (Fig EV4K). The impaired adhesion upon ORP2 removal agrees with earlier observations in ORP2 knockout hepatoma cells (Kentala *et al*, 2018).

### Acute ORP2 removal impairs cholesterol transfer from late endosomal organelles to integrin β1 recycling endosomes

To investigate endomembrane cholesterol distribution in acutely ORP2-depleted cells, we visualized intracellular cholesterol with

---

**Figure 3. FAK activity regulates cholesterol delivery to FAK/integrin endosomes and endomembrane PI(4,5)P$_2$ levels.**

A    After 1 day in 5% LPDS, NPC1-GFP expressing cells were loaded with 50 µg/ml LDL with or without 10 µM PF-228 for the indicated times, and stained for intracellular D4H. Representative confocal images of intracellular D4H, NPC1-GFP, and internalized integrin β1 labeling. Dashed lines indicate cell outlines.

B    Fraction of D4H-positive area in the cell as in (A).

C    Fraction of D4H area overlapping with NPC1 in a peripheral region toward the leading edge (exemplary box shown as inset in (A)).

D    Quantification of D4H area positive for integrin β1 in a peripheral region toward the leading edge. Mean ± SD, n = 26–30 cells pooled from 2 independent experiments. Student's *t*-test.

E–G    Control and FAK siRNA-treated cells were LDL depleted for 1 day, loaded with LDL for the indicated times, and stained for plasma membrane D4H or for intracellular D4H with lamp1 or integrin β1 antibodies. Quantification of plasma membrane D4H (E) and intracellular D4H area positive for lamp1 (F) or integrin β1 (G); corresponding images in Fig EV3D and E. Mean ± SD, n = 20–35 cells from 2 independent experiments. Student's *t*-test.

H    Schematic of FAK-regulated PI(4,5)P$_2$ generation on endosomes via PIPKIγ, modified from (Nader *et al*, 2016).

I    After 1-day 5% LPDS, cells were loaded with 50 µg/ml LDL with or without 10 µM PF-228 for the indicated times and endomembrane PI(4,5)P$_2$ was stained with antibody and imaged by confocal microscopy.

J    Quantification of mean PI(4,5)P$_2$ immunoreactivity in cells. Mean ± SD, n = 16–26 cells pooled from 2 independent experiments. Student's *t*-test.

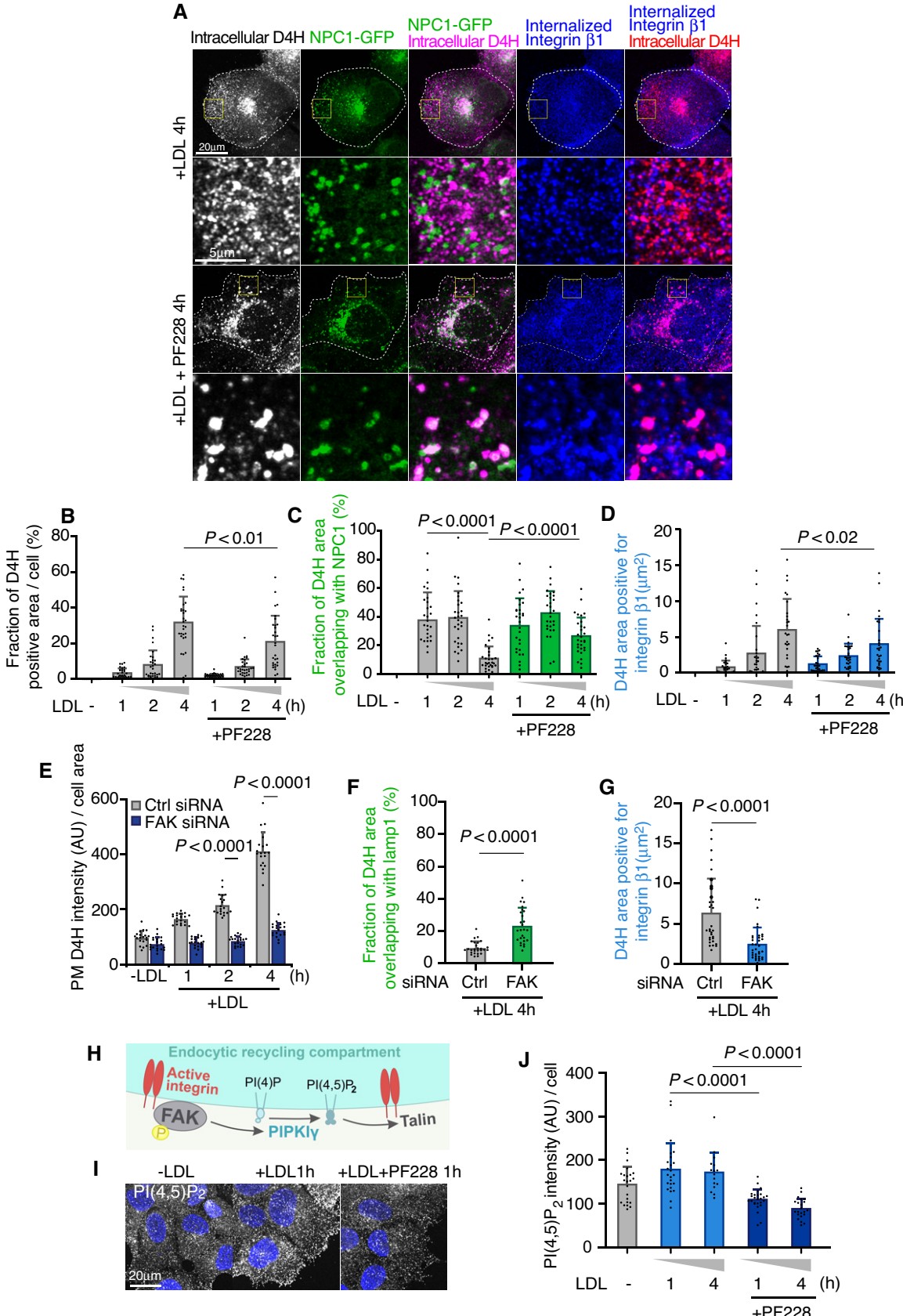

**Figure 3.**

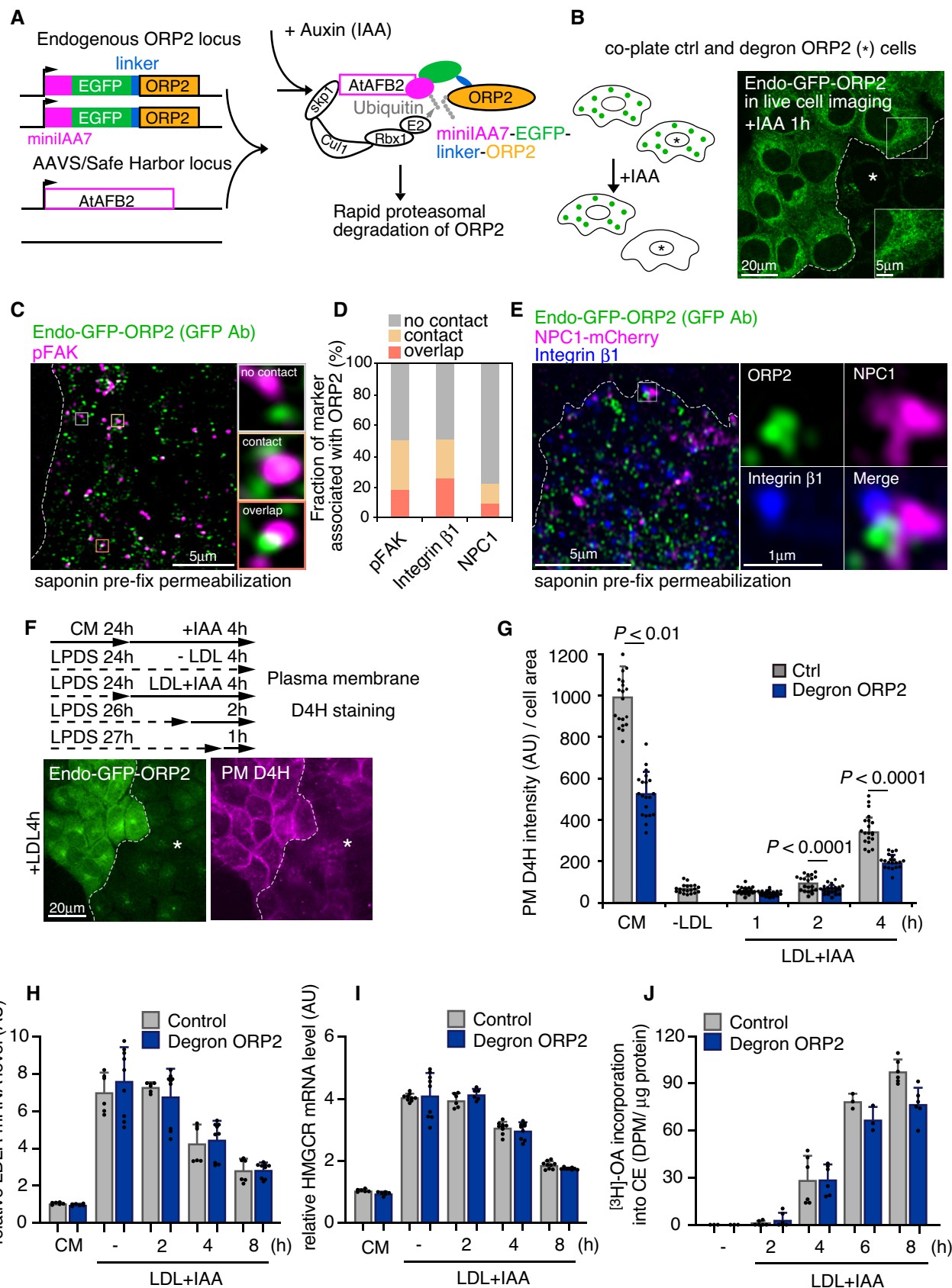

**Figure 4.**

**Figure 4.   Endogenous ORP2 localization and acute depletion via auxin-inducible degradation.**

A    Schematic of endogenous ORP2 tagging and auxin-inducible degron system.
B    Control (endogenously GFP-tagged ORP2 without AtAFB2) and degron-ORP2 cells were co-plated and treated with IAA for 1 h followed by live-cell confocal imaging. Dotted line separates GFP-ORP2 containing and depleted cells, asterisk indicates the area of ORP2-depleted cells, and inset shows no prominent PM enrichment of endo-GFP-ORP2.
C    Confocal image of endo-GFP-ORP2 and pFAK immunofluorescence staining in saponin pre-permeabilized cells loaded with LDL for 2 h before fixation. GFP antibody was used to enhance endo-GFP-ORP2 signal. Insets show examples of "no contact", "contact", and "overlap" of organelles. Dashed line indicates cell edge.
D    Quantification of pFAK, integrin β1, and NPC1 association with endo-GFP-ORP2 in C. $n$ = 380 pFAK; 353 integrin β1; 232 NPC1 organelles in 2 cells.
E    Representative confocal images of triple labeling for endogenous ORP2, NPC1-mCherry, and integrin β1 after 2 h of LDL loading, and saponin permeabilization prior to fixation. Dashed line indicates cell edge.
F    Schematic outline for plasma membrane D4H labeling in combination with ORP2-degron system, and representative widefield epifluorescent images of co-plated cells after 4 h of LDL loading. Asterisk indicates ORP2-depleted degron cells. Dashed lines separate ORP2 expressing and depleted cells.
G    Quantification of F. Mean ± SD, $n$ = 20 fields of cells pooled from 2 independent experiments. Student's $t$-test.
H–J   Control and degron-GFP-ORP2 cells were incubated in 5% LPDS for 1 day, loaded with LDL + IAA for the indicated times, and LDLR (H) and hydroxymethylglutaryl-CoA reductase (HMGCR) (I) mRNA levels were quantified with qPCR or [$^3$H]-oleic acid (OA) incorporation into cholesteryl esters (CE) was analyzed (J). Mean ± SD, $n$ = 2–3 RNA samples from independent experiments with triplicate measurements (H, I); $n$ = 3 (−LDL and 6 h); $n$ = 6 (all other time points) pooled from 2 independent experiments (J).

D4H as in Fig 2C. Control and degron-ORP2 cells were co-plated and, after LDL depletion, treated for increasing times with LDL and IAA. This revealed that the LDL-dependent increase in intracellular accessible cholesterol was attenuated when ORP2 was missing (Fig 5A and B). Antibody co-labelings revealed that the LDL-derived accessible cholesterol signal was initially confined largely to Lamp1-positive late endosomal/lysosomal organelles and that this fraction decreased over time efficiently in control but not in ORP2-depleted cells (Fig 5C and D; late endosomal compartments were detected with Lamp1 rather than NPC1 antibodies, because the latter were not compatible with liquid N$_2$ permeabilization). In parallel, the increase of D4H signal in integrin β1 compartments was not taking place in ORP2-depleted cells (Fig 5E and F). To address whether ORP2 modifies transient contacts between late and recycling compartments, we carried out live-cell imaging of contacts between dextran and transferrin-labeled endosomes before and after acute ORP2 removal (BODIPY-cholesterol imaging in GFP-ORP2 cells was not feasible due to overlapping fluorophores). This revealed that ORP2 removal for 2–3 h led to fewer but prolonged contacts between late and recycling compartments (Fig EV5A–C, Movies EV2 and EV3). Together, these data suggest that ORP2 controls the redistribution of LDL-cholesterol from late endosomal compartments to integrin-containing recycling endosomal compartments.

### ORP2 regulates endomembrane PI(4,5)P$_2$ and NPC1 distribution in a FAK-dependent manner

Considering that ORP2 can function as a cholesterol and PI(4,5)P$_2$ exchanger, we next investigated endomembrane PI(4,5)P$_2$ distribution and its relationship to ORP2. Endosomal PI(4,5)P$_2$ levels can be increased by silencing of the inositol polyphosphate 5-phosphatase OCRL (Vicinanza *et al*, 2011), as readily evident by the anti-PI(4,5)P$_2$ antibody labeling (Fig 6A). OCRL-deficient cells accumulate PI(4,5)P$_2$ both at early (Vicinanza *et al*, 2011) and late endocytic organelles (De Leo *et al*, 2016). Remarkably, in OCRL-silenced cells also endogenous GFP-ORP2 was redistributed to the perinuclear region, showing a pronounced colocalization with the PI(4,5)P$_2$ labeling (Fig 6A). This result indicates that endogenously expressed GFP-ORP2 can be recruited to PI(4,5)P$_2$ harboring endomembranes. LDL loading of cells caused a moderate increase in endomembrane PI(4,5)P$_2$ labeling but the pattern was more scattered than upon OCRL silencing (Fig EV5D and E). In contrast, in acutely ORP2-depleted

cells, LDL loading failed to increase the PI(4,5)P$_2$ labeling (Fig EV5D and E), suggesting that ORP2 is needed for the LDL-induced endomembrane PI(4,5)P$_2$ increase.

This prompted us to study the relationship between PI(4,5)P$_2$ and NPC1 more closely. In OCRL-silenced cells, some of the perinuclear PI(4,5)P$_2$ structures colocalized with NPC1-mCherry containing compartments (Fig 6B and C). Remarkably, overexpression of GFP-ORP2 resulted in enhanced anti-PI(4,5)P$_2$ labeling, with prominent colocalization with NPC1 and peripheral redistribution of the double labeled organelles (Fig 6B). This effect of ORP2 was dependent on its PI(4,5)P$_2$ binding capability, as overexpression of the PI(4,5)P$_2$ binding-deficient mutant (ORP2-mHHK) failed to increase the PI(4,5)P$_2$ labeling in NPC1 organelles (Fig 6B and C). Moreover, FAK activity was also involved, as its inhibition attenuated the ORP2-WT-induced increase of PI(4,5)P$_2$ in these organelles (Fig 6B and C).

To further investigate the effect of FAK and ORP2 on NPC1 organelles, we performed live-cell imaging of NPC1-mCherry expressing cells. The NPC1-containing structures are known to undergo extensive tubulation (Zhang *et al*, 2001); the tubulation, splitting, and kiss-and-run events enable fission and exit of cargo from lysosomes and related organelles (Saffi & Botelho, 2019). We found that FAK inhibition exaggerated the tubulation of NPC1-mCherry containing structures, as assessed by measuring the length of the longest tubule per cell (Fig 6D and E and Movies EV4 and EV5). Despite this, the delivery of NPC1 organelles to the cell periphery adjacent to FAs (Kanerva *et al*, 2013) was defective (Fig EV5F and G), arguing that the tubules may represent arrested carrier intermediates. The excessive tubulation of NPC1 organelles upon FAK inhibition could be rescued by overexpression of WT but not PI(4,5)P$_2$-binding-deficient mutant of ORP2 (Figs 6E and EV5H). Moreover, also when endogenous ORP2 was acutely depleted from NPC1-mCherry expressing cells, the NPC1 tubules became more elongated (Fig 6D and E). These results suggest that FAK activity and ORP2 with its PI(4,5,)P$_2$ transfer activity are needed to prevent excessive appearance of tubules in NPC1 organelles.

### Cholesterol- and PI(4,5)P$_2$ transfer-competent ORP2 enhances FAK activation

Finally, since PI(4,5)P$_2$ regulates the membrane association and activation of FAK (Goni *et al*, 2014; Bauer *et al*, 2019; Acebron *et al*,

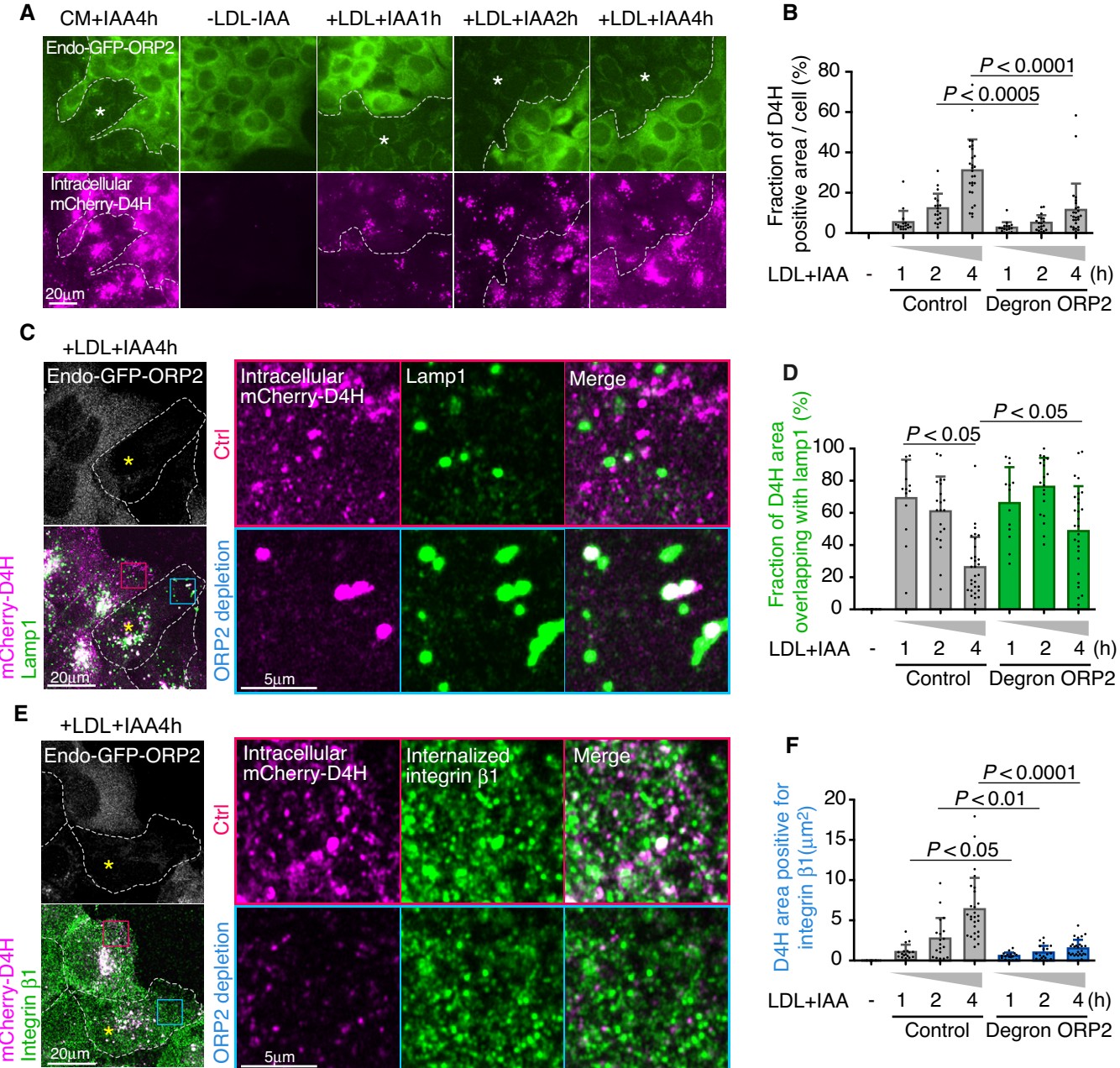

**Figure 5. Acute removal of endogenous ORP2 impairs LDL-cholesterol transfer from late endosomes to FAK/integrin endosomes.**

A    Control and degron-ORP2 cells were co-plated, and after 1-day incubation in 5% LPDS, cells were loaded with LDL (+LDL) and IAA (+IAA; ORP2 depletion) for the indicated times. Intracellular cholesterol was labeled with D4H and imaged by widefield epifluorescence microscopy. Asterisk indicates region of ORP2-depleted cells. CM, complete medium. Dashed lines separate ORP2 expressing and depleted cells.

B    Quantification of (A). Mean ± SD, n = 15–27 cells pooled from 2 independent experiments. Student's t-test.

C–F  Control and degron-ORP2 cells were co-plated, and after 1 day in 5% LPDS, treated with 50 µg/ml LDL and IAA for the indicated times. Asterisk indicates an ORP2-depleted cell. Double labeling of intracellular D4H and Lamp1 (C) or intracellular D4H and internalized integrin β1 (E). Images were acquired by confocal microscopy. Red box indicates a peripheral region toward the leading edge in the control cell, and blue box a corresponding region in the ORP2-depleted cell. (D, F) Fraction of D4H area overlapping with lamp1 (D) and D4H area positive for internalized integrin β1 (F) in a peripheral region toward the leading edge (exemplary box as inset). Mean ± SD, n = 20–30 cells pooled from 2 independent experiments. Student's t-test.

2020) we studied whether ORP2 plays a role in FAK membrane association and phosphorylation. LDL loading increased the pFAK immunostaining of cells in line with the immunoblotting results (Fig 1F), displaying a prominent intracellular punctate pattern, but

this effect was missing in acutely ORP2-depleted cells (Figs 7A and B, and EV5I and J). The pFAK labeling was rescued by introduction of wild-type ORP2 but not by the PI(4,5)P$_2$- or sterol binding-deficient mutants (Figs 7A and B, and EV5K). These data suggest

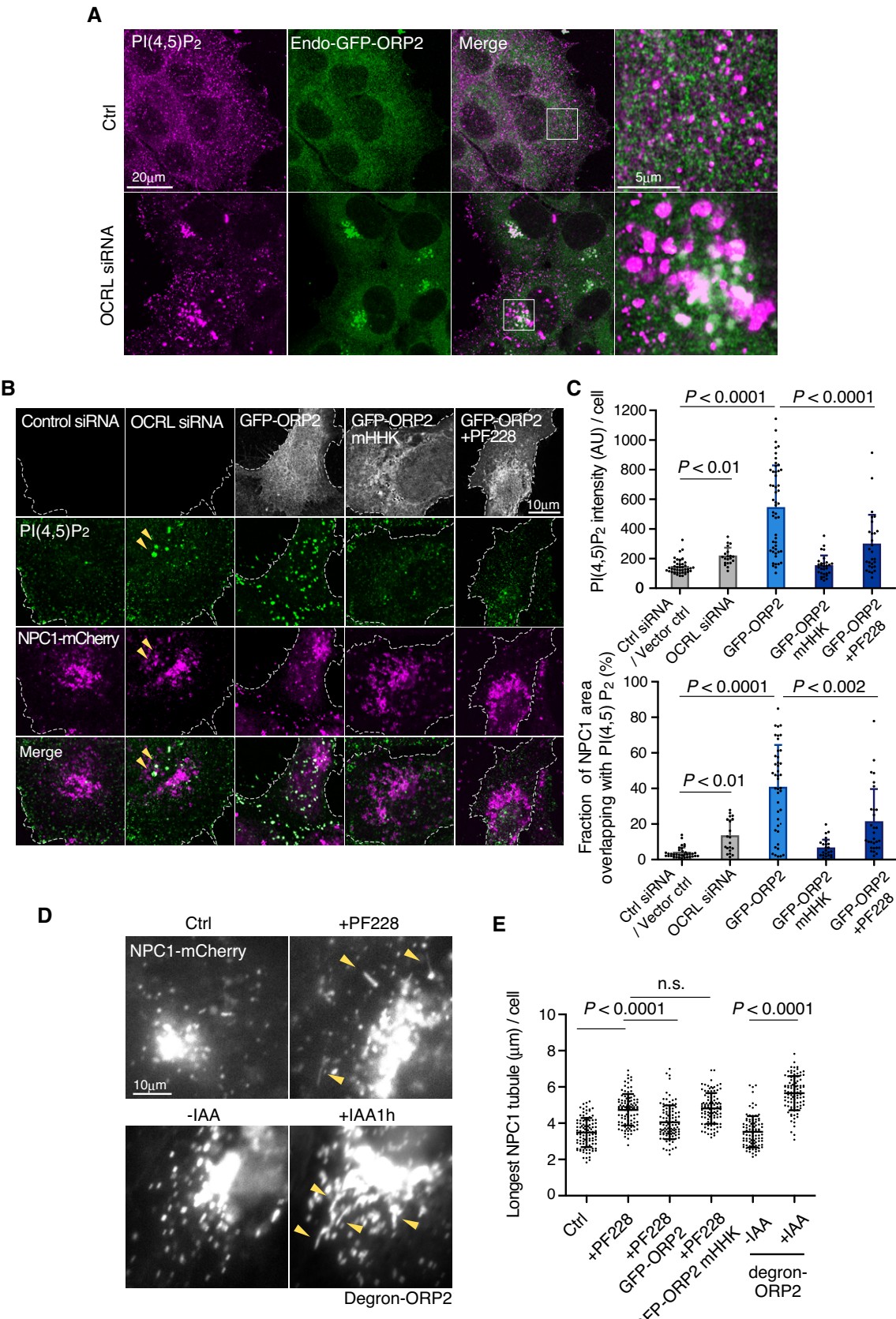

**Figure 6.**

**Figure 6. ORP2 regulates PI(4,5)P$_2$ and NPC1 distribution.**

A   Endo-GFP-ORP2 cells were treated with control or OCRL siRNAs for 2 days, stained with PI(4,5)P$_2$ antibody, and imaged by confocal microscopy.
B   Cells stably expressing NPC1-mCherry were treated with control or OCRL siRNAs for 2 days, or transfected with GFP-ORP2 or -ORP2-mHHK (PI(4,5)P$_2$ binding-deficient mutant) for 1 day, stained with PI(4,5)P$_2$ antibody, and imaged by confocal microscopy. For FAK inhibition, GFP-ORP2 transfected cells were treated with 10 μM PF-228 for 4 h. Dashed lines indicate cell edges. Arrowheads indicate colocalization of PI(4,5)P$_2$ and NPC1.
C   Quantification of (B). Values from control siRNA and Vector control were pooled and are shown as control siRNA/Vector control. Mean ± SD, $n$ = 20–43 cells pooled from 2 independent experiments. Student's $t$-test.
D, E   Cells stably expressing NPC1-mCherry were transfected with GFP-ORP2 or GFP-ORP2-mHHK for 1 day and were indicated, treated with 10 μM PF-228 for 4 h. For degron-ORP2 cells, NPC1-mCherry was transiently transfected for 2 days and cells incubated without (-IAA; ORP2 present) or with IAA (+IAA; ORP2 depleted) for 1 h. Arrowheads indicate tubular NPC1 organelles. Live cell images were acquired by widefield epifluorescence miroscopy and the longest NPC1 tubule per cell was measured. Mean ± SD, $n$ = 102–122 cells. Students's $t$-test. Please see also control (Movie EV4) and PF-228-treated cell (Movie EV5), both videos 1 min recordings with 1 s frame rate.

that ORP2 regulates endomembrane PI(4,5)P$_2$ levels and FAK activation pending on its PI(4,5)P$_2$ and cholesterol transfer.

FAK activation is known to be initiated by binding of its FERM domain to PI(4,5)P$_2$-containing membranes (Goni *et al*, 2014; Bauer *et al*, 2019; Acebron *et al*, 2020). We therefore carried out biochemical fractionations of LDL-loaded control and acutely ORP2-ablated cells. This showed that ORP2 enhances the membrane association of FAK (Fig 7C). One possibility is that cholesterol, presumably transferred by ORP2, contributes to the binding of FAK FERM to PI(4,5)P$_2$-containing membranes. To address this step *in vitro*, we purified the FAK FERM domain and investigated its binding to liposomes with defined lipid compositions. In liposome cosedimentation assays (Fig 7D), the membrane binding of FAK FERM domain was increased by PI(4,5)P$_2$ in a concentration-dependent manner, as expected (Fig 7E and F). Addition of 20 mol% cholesterol to membranes containing 1–2 mol% PI(4,5)P$_2$ or 40 mol% cholesterol to membranes containing up to 5 mol% PI(4,5)P$_2$ further enhanced the membrane association of FERM (Fig 7E and F). As an alternative strategy, we labeled the purified FERM domain with Alexa488 and studied its binding to giant unilamellar vesicles (GUVs) by imaging (Fig 7G). Control experiments confirmed that the fluorescently labeled AF488-FERM binds to liposomes in a PI(4,5)P$_2$ concentration-dependent manner similarly as non-labeled FERM (Fig EV5L). Addition of 30 mol% cholesterol to PI(4,5)P$_2$-containing GUVs increased the FERM domain membrane binding by roughly twofold (Fig 7H). Together, these data argue that cholesterol facilitates the association of FAK with PI(4,5)P$_2$-containing membranes.

## Discussion

Here, we investigated the itinerary of LDL-derived cholesterol to the PM. LDL particle hydrolysis by acid lipase is initiated in early endosomes (Sugii *et al*, 2003) and NPC1, a key gatekeeper for LDL-cholesterol export, localizes to late endocytic organelles (Ko *et al*, 2001; Zhang *et al*, 2001) that communicate extensively with other endomembranes. We found that rapidly after becoming accessible in NPC1 organelles (after 1–2 h of LDL loading), LDL-cholesterol becomes accessible in integrin recycling organelles. As the NPC1 and integrin recycling compartments do not mix, this implies that LDL-cholesterol is transferred to integrin recycling organelles, which also harbor the key adhesion regulator FAK.

Our results argue that cholesterol transfer from the NPC1 organelles to recycling compartments is regulated by ORP2 (see Fig 8), as upon acute degradation of endogenous ORP2, LDL-cholesterol does not become accessible in integrin recycling compartments and FAK activation is attenuated. Importantly, ORP2 increases the membrane association of FAK in LDL-loaded cells and cholesterol enhances the membrane association of FAK FERM domain, a prerequisite for FAK activation. Active FAK promotes PI(4,5)P$_2$ generation in the endomembranes via activation of PIPKIγ (Nader *et al*, 2016). Our data further reveal that endogenous ORP2 has affinity for endosomal PI(4,5)P$_2$ and that ORP2 can promote the backtransfer of PI(4,5)P$_2$ to NPC1 organelles. Which proteins this late endosomal PI(4,5)P$_2$ interacts with remains to be elucidated but, based on our findings, it is likely involved in assembling machineries for late endosomal dynamics and cargo export. Indeed, late endosomal/lysosomal PI(4,5)P$_2$ is considered to regulate kinesin-driven

**Figure 7. Cholesterol dependent activation of FAK.**

A, B   Control and degron-ORP2 cells were co-plated, and after 1-day 5% LPDS, treated with 50 μg/ml LDL and IAA (+IAA; ORP2 depletion) for the indicated times and immunostained for pFAK to quantify its intensity. For rescue experiments, degron-ORP2 cells were plated and transfected with GFP-ORP2 or -ORP2-mHHK or -ORP2-ΔELSK for 6 h. Cells expressing GFP-ORP2 constructs at levels similar to endo-GFP-ORP2 were used for quantifying pFAK intensity. Dashed lines indicate cell outlines. See also Fig EV6C. Asterisk indicates cells depleted of endogenous ORP2. Images were acquired by confocal microscopy. Mean ± SD, $n$ = 20–23 cells pooled from 2 independent experiments. Student's $t$-test.
C   Quantification of FAK protein in cytosolic and membrane fractions in degron-ORP2 cells with or without IAA. Cells were starved overnight in LPDS and loaded with 50 μg/mL LDL +/− IAA for 2 h. The proportion of FAK signal in the membrane fraction (of total FAK in cytosol + membranes) is presented ± SD. For no IAA $n$ = 13 and for IAA $n$ = 11, 2 independent experiments. Student's $t$-test.
D–F   Liposome-co-sedimentation of FAK FERM with increasing concentrations of PI(4,5)P$_2$ and cholesterol. "S" in (D) indicates the supernatant and "P" pellet containing the FAK FERM-bound to liposomes. The total lipid concentration was kept at 50 μM, and purified FAK FERM concentration at 1 μM. The initial lipid composition was POPC:POPE:POPS:Rhodamine-DHPE:PI(4,5)P$_2$:cholesterol (70:19:10:1:0:0, mol/mol). The increasing PI(4,5)P$_2$ and cholesterol concentrations were compensated by decreasing the amount of POPC. Mean ± SD, $n$ = 5. Student's $t$-test (*$P$ < 0.05; **$P$ < 0.002). Numbers under the blots indicate fraction of FAK FERM bound to liposomes (pellet) of total FAK FERM (supernatant + pellet).
G, H   Binding of Alexa Fluor 488-labeled FAK FERM domain to GUVs containing PI(4,5)P$_2$ and cholesterol. The lipid compositions were POPC:POPE:POPS:Rhodamine-DHPE:PI(4,5)P$_2$:cholesterol (50:19:10:1:20:0) and POPC:POPE:POPS:Rhodamine-DHPE:PI(4,5)P$_2$:cholesterol (20:19:10:1:20:30, mol/mol). Images were acquired by widefield epifluorescence microscopy. Scale bar, 5 μm. Mean ± SD, $n$ = 68–76 pooled from 2 independent experiments. Student's $t$-test.

Source data are available online for this figure.

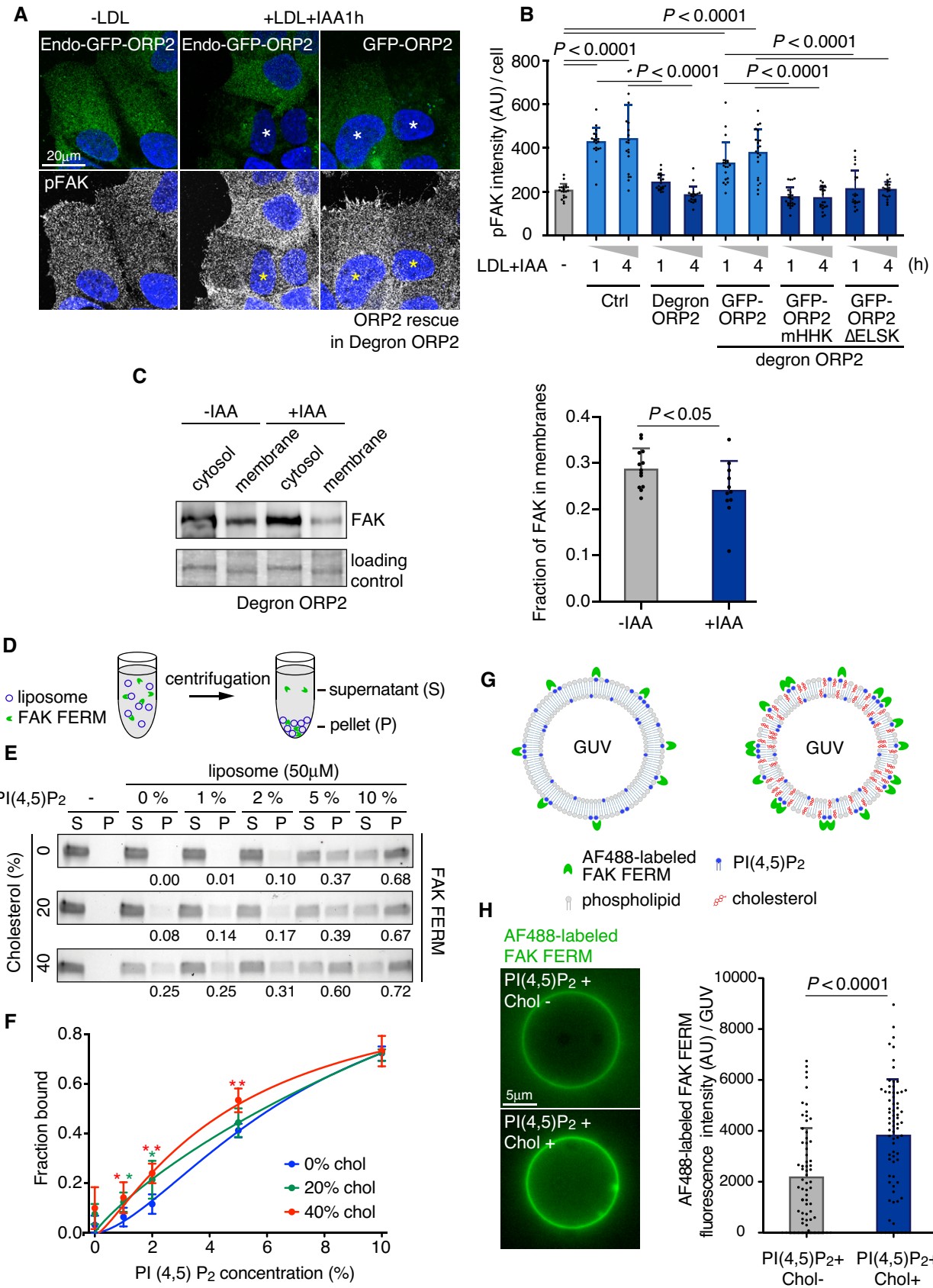

**Figure 7.**

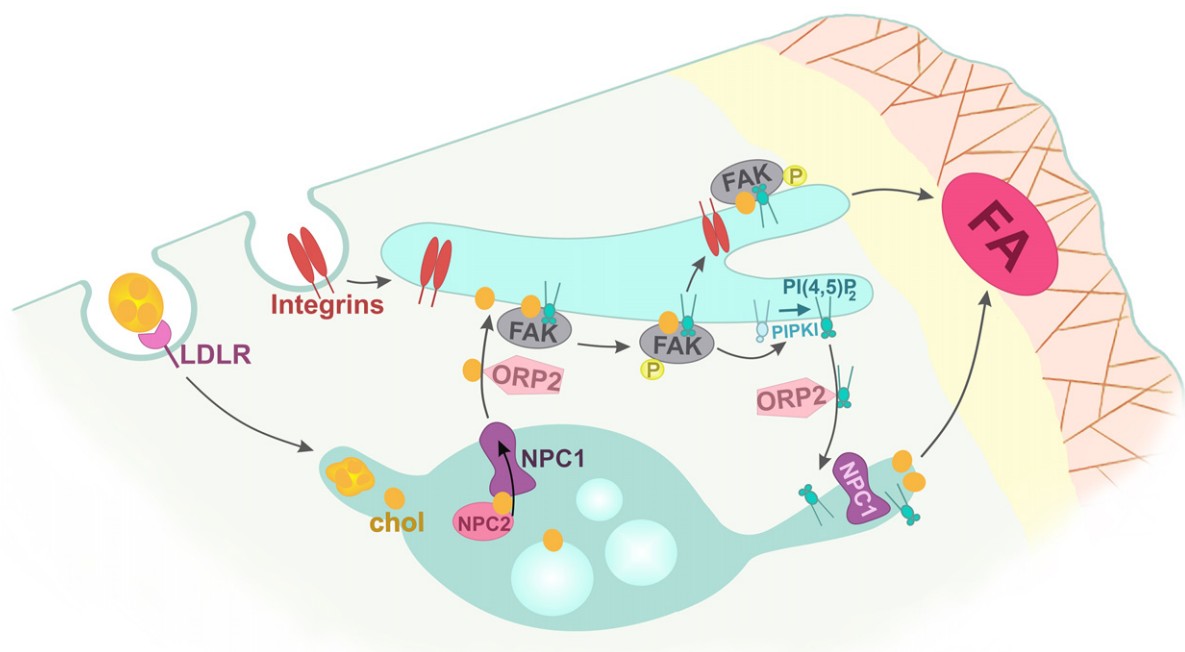

**Figure 8.  Schematic of LDL-cholesterol delivery to the PM.**

LDL-cholesterol is taken up via LDL receptor-mediated endocytosis and the liberated free cholesterol is incorporated into late endosomal limiting membrane via NPC2 and NPC1. ORP2 delivers cholesterol from late endosomes to FAK/integrin recycling endosomes where cholesterol facilitates FAK association with the PI(4,5)P$_2$-containing membrane. Activated FAK increases the activity of endosomal PIPKIγ which generates PI(4,5)P$_2$, accelerating the recycling of active integrins. ORP2 unloads PI(4,5)P$_2$ from FAK/integrin endosomes and delivers it to NPC1-containing late endosomes, regulating their tubulovesicular dynamics. This coupling of ORP2 and FAK activity drives efficient cholesterol delivery to the plasma membrane, stimulating FA dynamics and cell migration.

organelle movement and nucleation of lysosomal scission machinery (Ebner *et al*, 2019; Saffi & Botelho, 2019).

The present study employed for the first time rapid (mins-to-hrs) protein perturbation to investigate the function of endogenous ORP2. This is relevant in lipid trafficking/metabolism where the removal of a single gene product can be accompanied by complex adaptive changes. For instance, inducible ORP2 overexpression for 2 days enhanced cholesterol efflux to extracellular acceptors, decreased cellular cholesterol, and resulted in upregulation of cholesterol synthesis and LDL uptake (Hynynen *et al*, 2005). We also detected ORP2 ligands using strategies that *per se* do not affect lipid trafficking/distribution (as e.g., intracellular overexpression of lipid-binding protein domains might do): Accessible cholesterol was detected with the biosensor D4H applied extracellularly and PI(4,5)P$_2$ by antibodies only post, not during the process under study (e.g., LDL loading, FAK inhibition). Of note, in most experiments LDL was loaded into cells for increasing times. Hence, the NPC1 compartment is not getting emptied of LDL-cholesterol and the relative contribution of NPC1 vs. FAK/integrin endosomes in LDL-cholesterol PM delivery cannot be specified. This may also vary between cells: e.g., in motile cells, where the turnover of adhesions is accelerated, the integrin recycling circuit route might play a more prominent role than in statically adherent cells.

We found endogenous GFP-ORP2 in the vicinity of NPC1 and integrin/FAK containing organelles. Tracing their communication with LDL-derived BODIPY-cholesterol revealed rapid kiss-and-run like endosome contacts, suggesting that ORP2-mediated lipid exchange occurs during such transient contacts. Long-term membrane

anchoring of ORP2 would not be needed in such a scenario (Wang *et al*, 2019), as opposed to, e.g., more stable ORP2 binding to the ER via VAPs (Weber-Boyvat *et al*, 2015). Rather, ORP2 function—presumably in lipid exchange, to govern the correct membrane lipid compositions—appears important for maintaining the transient nature of contacts between late and recycling endosomal compartments, as such contacts became fewer in number and longer-lasting rapidly upon ORP2 removal.

Based on structural information, the cholesterol-bound ORP2 is monomeric and the PI(4,5)P$_2$-bound one tetrameric (Wang *et al*, 2019), i.e., four cholesterol molecules would be transported from late to recycling organelles by ORP2 monomers that assemble to a tetramer to transport four PI(4,5)P$_2$ molecules back. Cholesterol transfer from LDL-loaded late endosomes to recycling endosomes is likely down the concentration gradient; hence, PI(4,5)P$_2$ hydrolysis would not be needed to energize it but PI(4,5)P$_2$ could instead be used for assembling endosomal PI(4,5)P$_2$-interacting proteins. Of note, our data do not conclusively rule out ORP2 function in cholesterol-PI(4,5)P$_2$ exchange between endosomes and the PM, but since endogenous ORP2 was not concentrated in the PM and a detectable PM enrichment of PI(4,5)P$_2$ only developed with a lag upon ORP2 removal, a primary lipid exchange function at the level of endosomes appears more likely. Instead, we found no evidence that LDL-cholesterol trafficking toward the ER would be perturbed immediately upon ORP2 degradation.

Our findings reveal a previously unsuspected role for FAK in controlling LDL-cholesterol delivery and regulating NPC1 organelle dynamics. If FAK is inactive, NPC1 organelles undergo excessive tubulation and

fail to reach the peripheral sites close to FAs. As overexpression of PI(4,5)P$_2$ transfer-competent (but not incompetent) ORP2 promotes peripheral scattering of NPC1 organelles and can bypass FAK activation in limiting NPC1 tubulation, PI(4,5)P$_2$ generation enhanced by FAK is probably one, if not the sole function of FAK in LDL-cholesterol delivery. Overexpressed ORP2 can probably transfer PI(4,5)P$_2$ independently of endosomal PIPKIγ activity, for instance from the PM, and thereby circumvent the need for FAK. Physiologically, ORP2 cross-talks with FAK, with ORP2-mediated cholesterol transfer stimulating FAK membrane association and activation, and FAK activity promoting PI(4,5)P$_2$ generation for ORP2-mediated transfer. This cross-talk may be relevant for the *in vivo* function of ORP2: The mechanosensory bundles of auditory cells are composed of cilia, which depend on FAK (Antoniades *et al*, 2014), ORP2 knockout pigs had degenerated stereocilia (Yao *et al*, 2019) and ORP2-deficient cochlear cells displayed inhibited FAK activity (Shi *et al*, 2020). To conclude, the present study pinpoints the activity of ORP2 in cholesterol-PI(4,5)P$_2$ transfer between late and recycling endosomes, in concert with a new function of FAK in controlling late endosomal trafficking and cholesterol egress.

# Materials and Methods

## Materials

Human LDL was prepared from pooled plasma of healthy volunteers by sequential ultracentrifugation, and lipoprotein-deficient serum was prepared as described in (Goldstein *et al*, 1983). DiI-LDL was prepared as described in (Reynolds & St Clair, 1985). BODIPY-cholesteryl linoleate was synthesized and used to label human LDL as described (Kanerva *et al*, 2013). Methyl-β-cyclodextrin (C4555), PF-228 (PZ0117) and U18666A (C3633) were obtained from Sigma. Indole-3-acetic acid sodium (IAA, Santa Cruz, sc-215171) was prepared at 100× in H$_2$O (10 mg/ml), aliquoted, stored at −20°C and used within 2 days after thawing. Antibodies: pFAK (BD Transduction Laboratory #611807, clone 18 for Western blotting and Thermo Fisher #44-624G for immunofluorescence), FAK (Sigma 05-537, Clone 4.47), integrin β1 (Santa Cruz sc-18887, K20 for immunofluorescence and Sigma MAB2252, clone N29 for Western blotting), PI(4,5)P$_2$ (Echelon, Z-G045, clone 2C11), ORP2 (Novus Biologicals, NBP1-92236), NPC1 (Abcam, ab134113), Lamp1 (DSHB, H4A3), GFP (Abcam, ab290 and ab1218); see Appendix Fig S1 for specificity controls of pFAK, FAK, and integrin β1antibodies. Lipids: 1-palmitoyl-2oleoyl-sn- glycero-3-phosphocholine (POPC), 1-palmitoyl-2oleoyl-sn-glycero-3-phosphoethanolamine (POPE), 1-palmitoyl-2oleoyl-sn-glycero-3-phospho-L-serine (sodium salt) (POPS), L-a-phosphatidylinositol-4,5-bisphosphate (brain, porcine, ammonium salt; brain PI(4,5)P$_2$), cholesterol (ovine) were from Avanti Polar Lipids. Lisssamine™ rhodamine B 1,2-dihexadecanoyl-sg-glycero-3-phosphoethanolamine, triethyllammonium salt (rhodamine-DHPE) was from ThermoFisher Scientific. PI(4,5)P$_2$ was dissolved in chroloform:methanol:H$_2$O (20:9:1) and other lipids in choroloform.

## Cell culture and transfection

Human epithelial carcinoma A431cells (ATCC, Cat#CRL-1555) were maintained in Dulbecco's modified Eagle's medium (Lonza) containing 10% FBS (Gibco), penicillin/streptomycin (100 U/ml each,

Lonza), and 2 mM L-glutamine (Gibco) (complete medium, CM) at 37°C in 5% CO$_2$. For LDL depletion, cells were incubated in DMEM supplemented with 5% LPDS prepared as described in (Goldstein *et al*, 1983) for ~ 1 day, and 50 μg/ml LDL was loaded for the indicated times. Transfections were performed with X-tremeGENE HD Transfection Reagent (Sigma) for plasmids and HiPerFect (Qiagen) for siRNAs according to the manufacture's protocol.

## Plasmids and siRNAs

The following plasmids were obtained from Addgene: mCherry-talin (Addgene plasmid #80026) (Lee *et al*, 2013), mCherry-FAK (#55122), GFP-FAK (#50515) (Gu *et al*, 1999), GFP-FAK Y397F (#50516), and iRFP-PH-PLCdelta1 (#66841) (Idevall-Hagren *et al*, 2012). GFP-paxillin was provided by Dr Pekka Lappalainen (University of Helsinki). GFP or mCherry was linked to the C-terminus of NPC1 and cloned into the safe harbor vector pSH-FIRE-P-AtAFB2 (Addgene, #129715) (Li *et al*, 2019) to obtain pSH-NPC1-GFP or -mCherry. GFP-ORP2, phosphoinositide binding-deficient (mHHK: H$^{178-179}$A, K$^{423}$A), and sterol nonbinding (ΔELSK: deletion of residues 430–433) mutant in pEGFP-C1 were described in REF (Weber-Boyvat *et al*, 2015). mCherry-D4H cDNA in mCherry-C1 vector was provided by Dr Gregory G. Fairn (University of Toronto) (Maekawa & Fairn, 2015). The D4H domain was PCR-amplified and inserted into pGEX6P-1 with N-terminal sfGFP- or mCherry-tag. Human FAK FERM domain (34–362) was PCR-amplified from mCherry-FAK incorporating EcoRI and XhoI restriction sites and inserted into pGEX6P-1 (GE Life Sciences). The following siRNAs (from Ambion) were used: silencer select negative control #1, FAK siRNA (GAUGUUGGUUUAAAGCGAUtt, ID: s11485) integrin β1 siRNA (CCGUAGCAAAGGAACAGCAtt, ID: 7574), OCRL siRNA (GGGUCUCAUCAAACAUAUCtt, ID: 104450), and ORP2 siRNA (CACUCUUGGGGAGAAACGUAtt, ID: s19149).

## Protein purification

pGEX6P-sfGFP/mCherry-D4H was transformed into BL21 (DE3) competent cells, and protein expression was induced with 0.4 mM isopropyl-β-D-thiogalactopyranoside (IPTG, Sigma) at 16°C overnight. The cell pellet was lysed and sonicated in lysis buffer (50 mM NaH$_2$PO$_4$/Na$_2$HPO$_4$, 300 mM NaCl, pH 8.0) complemented with 1 mM benzamidine, 10 μg/ml chymostatin, 10 μg/ml leupeptin, 2 μg/ml pepstatin A, and 0.4 mM PMSF (Sigma). The lysate was centrifuged at 12,000 *g* for 10 min at 4°C. The supernatant was incubated with pre-equilibrated Glutathione Sepharose 4B beads (GE Life Sciences) for 2 h at 4°C. The resin was washed with lysis buffer, the GST-tag was removed with PreScission protease (GE Life Sciences), and sfGFP- or mCherry-D4H was eluted with elution buffer (20 mM NaH$_2$PO$_4$/Na$_2$HPO$_4$ pH 7.4). pGEX6P-1-FAK FERM was transformed into BL21 (DE3) competent cells, and protein expression was induced with 0.2 mM IPTG at 18°C overnight. The cell pellet was lysed and sonicated with lysis buffer A (50 mM Tris–HCl, pH 8.0, 250 mM NaCl) complemented with 1 mM EDTA, 1 mM TCEP, 1 mM benzamidine, 10 μg/ml leupeptin, 1 mM PMSF, 10 mg/ml lysozyme, and 0.5 mg/ml DNase. The lysate was treated with 1% Triton X-100 for 15 min at 4°C and centrifuged at 12,000 *g* for 10 min. The supernatant was incubated with pre-equilibrated Gluthatione Sepharose 4B beads in buffer B (50 mM Tris–HCl, pH 7.0, 250 mM NaCl, 1 mM EDTA, 1 mM TCEP and 1% Triton X-100) for 2 h at 4°C. The resin

was washed with buffer B, the GST-tag was removed with PreScission protease, and FAK FERM was eluted with elution buffer (50 mM Tris–HCl, pH 7.4, 150 mM NaCl, 1 mM EDTA, and 1 mM TCEP). For fluorescent labeling, ten-times molar excess of Alexa 488 maleimide (Life Technologies) was added to react with cysteines overnight at 4°C. Excess dye was removed by dialysis (MWCO 6-8000) in 20 mM Tris–HCl, pH 7.5, 150 mM NaCl, and 1 mM EDTA. Purified D4H and FAK FERM were stored at −20 and −80°C, respectively.

### D4H staining of cells

For plasma membrane D4H staining, cells were grown on glass cover slips, incubated in sfGFP- or mCherry-D4H solution (10 μg/ml in PBS, 1% BSA) for 30 min on ice, washed with ice-cold PBS once, and fixed with 4% PFA for 15 min. Cover slips were rinsed in $H_2O$ and mounted on slides using Mowiol/DABCO (Calbiochem/Sigma). For intracellular D4H staining, cells were grown on glass cover slips, fixed with 4% PFA, permeabilized in a liquid nitrogen bath for 30 s as described (Wilhelm *et al*, 2017), and incubated with D4H solution (10 μg/ml in PBS, 1% BSA) at RT for 1 h. Cells were then washed with PBS, re-fixed with 4% PFA for 15 min, and blocked with 1% BSA in PBS at RT for 30 min. Primary antibodies were incubated at RT for 1 h followed by incubation with Alexa Fluor-conjugated secondary antibodies at RT for 1 h.

### Quantification of FA dynamics

To analyze FA dynamics, cells were transfected with GFP-paxillin, mCherry-talin, GFP-FAK, or GFP-FAK Y397F plasmids and plated on fibronectin-coated (5 μg/ml) glass bottom μ–slide 4-well plates (Ibidi), to promote integrin-dependent cell adhesion mechanisms. Single cells were chosen and images acquired every 30 s for 1–2 h, using a Nikon eclipse-Ti-E N-STORM microscope, equipped with Andor iXon+ 897 back-illuminated EMCCD camera and 100× Apo TIRF oil objective, NA 1.49. For LDL loading, cells were LDL depleted in 5% LPDS for 1 day, and incubated with 50 μg/mL LDL in DMEM for the indicated time points. FA assembly and disassembly rates were analyzed by uploading the videos to FA analysis server (FAAS; http://faas.bme.unc.edu/) using the following settings: detection threshold, 2.5; Min adhesion size, 2; Phase length, 5 images; Min FAAI ratio, 3. For NPC1 and FAK kinase inhibition, 2 μg/mL U18666A or 10 μM PF-228 was added after 30 min of LDL loading. For cholesterol depletion, cells were treated with 5 mM methyl-β-cyclodextrin for 30 min. Unless otherwise noted, images were acquired during 1–3 h of LDL loading.

### Fixed and live-cell widefield epifluorescence imaging

Cells were imaged with Nikon eclipse-Ti-E microscope equipped with Nikon Perfect Focus System 3, Hamamatsu Flash 4.0 V2 scientific CMOS and Okolab stage top incubator system. For plasma membrane D4H imaging, 40× Fluor oil objective NA 0.75 was used and fields of cells were randomly acquired. Imaging was carried out at ~ 60–70% cell confluency and mean fluorescence intensity of non-cell region in the same image was used for background subtraction. For live-cell imaging (Figs 6D and EV5H), 100× Plan Apo VC oil objective was used. Cells were seeded onto 1 μ-slide 8-well ibiTreat chambers coated with 5 μg/ml fibronectin and imaging was

performed at 37°C, 5% $CO_2$. For the movies, cells were imaged with 1 s frame rate for 1 min.

### Fixed and live-cell confocal imaging

All images were acquired at ~ 60–70% cell confluency, and cells migrating toward the free space were chosen for imaging. Cells were imaged with Zeiss LSM 880 equipped with an Airyscan detector using a 63× Plan Apochromat oil objective NA 1.4, with sequential excitation of fluorophores using appropriate lasers and stable emission filter sets. The Airyscan detector was adjusted regularly between acquisitions. Images were Airyscan processed automatically using the Zeiss Zen software package. For intracellular D4H quantification, fraction of D4H area overlapping with other organelles in a peripheral region toward the leading edge per cell was analyzed (to avoid the perinuclearly clustered signal). Images were thresholded, and quantified using Mander's overlap coefficient as a measure of colocalization. For PI $(4,5)P_2$ quantification (Fig 6B and C), mean fluorescence intensity or fraction of NPC1 area overlapping with PI$(4,5)P_2$ per cell was analyzed. Imaging was done within 24 h after fixation. For live-cell imaging, cells were seeded onto 8-well Lab-Tek II #1.5 coverglass slides coated with 5 μg/ml fibronectin and imaging was performed at 37°C, 5% $CO_2$. Medium was replaced with FluoroBrite DMEM (Thermo Fisher Scientific) before live-cell imaging. For plasma membrane PI $(4,5)P_2$ imaging, cells were transfected with iRFP-PH-PLCdelta1 for overnight and imaged live.

### BODIPY-cholesteryl linoleate/LDL pulse-chase and imaging

Cells seeded on fibronectin-coated 4-well Lab-Tek II #1.5 coverglass slides were grown overnight in 5% LPDS medium supplemented with 50 μg/ml Alexa Fluor 568-dextran (Thermo Fisher Scientific) as indicated and then pulse-labeled with 50 μg/ml BODIPY-cholesteryl linoleate/LDL in serum-free DMEM for 2 h as described (Kanerva *et al*, 2013). For detection of integrin or transferrin, cells were incubated with 2 μg/ml Alexa Fluor 680-conjugated integrin β1 antibody (K20, Santa Cruz, sc-18887) for 30 min or with 50 μg/ml Alexa Fluor 647-labeled human transferrin (Thermo Fisher Scientific) for 15 min before imaging. Cells were then washed and chased in serum-free $CO_2$-independent medium (Gibco) for the indicated times and imaged live at 37°C. Confocal imaging was performed on a Leica TCS SP8 X inverted microscope (Leica Microsystems) with ×63 HC PL APO CS2 water objective, NA 1.20. For time series, single focal planes were imaged with 145 ms frame rate for 1 min. The fraction of BODIPY-cholesterol residing in integrin β1-positive or transferrin organelles was quantified from deconvolved (SVI Huygens), background-subtracted images with ImageJ FIJI by using Mander's overlap coefficient as a measure of colocalization. The increase in organellar BODIPY-cholesterol intensity was analyzed by measuring BODIPY-cholesterol mean intensity in integrin β1-positive organelles either contacting or not contacting with other organelle/s in the same time series.

### Dextran and transferrin pulse-chase and imaging

Degron ORP2 cells seeded on Lab-Tek II slides were grown overnight in 5% LPDS, serum-starved for 1 h, and then incubated with 50 μg/ml LDL and 500 μg/ml fluorescein dextran (Thermo Fisher

Scientific, D1820) for 2 h. After washing, the cells were loaded for 15 min with 50 µg/ml Alexa Fluor 647-labeled human transferrin and washed again. IAA (100 µg/ml) was added simultaneously with LDL and kept on during the chase. The cells were chased and imaged as in BODIPY-cholesteryl linoleate/LDL experiments except that imaging was performed with 369 ms frame rate. Trajectories of double-positive organelles were traced and measured with ImageJ FIJI using Manual Tracking plugin. Dark pixels between organelles was considered as lack of contact. To measure the number of organelle contacts, individual dextran positive organelles were followed for 7 s and the number of transferrin organelles touching a dextran organelle during this time was quantified.

## Immunofluorescence microscopy

Cells were grown on coverslips, fixed with 4% PFA for 15 min, permeabilized with 0.1% Triton X-100 for 10 min and blocked with 1% BSA in PBS for 30 min at RT. Cells were then stained with primary antibody for 1 h at RT followed by secondary antibody for 1 h at RT. Cover slips were rinsed in $H_2O$ and mounted on slides using Prolong glass antifade reagent (Thermo Fisher Scientific). For detection of internalized integrin β1, cells were incubated with 2 µg/ml integrin β1 at 37°C for 30 min before fixation, permeabilized, and incubated with Alexa Fluor-conjugated secondary antibody. For pre-fixation permeabilization, cells were washed with DPBS and permeabilized with 0.0025% saponin in 10 mM MES, pH 6.1, containing 138 mM KCl, 3 mM $MgCl_2$, 2 mM EGTA, and 320 mM sucrose for 1 min at 37°C. Cells were gently washed with DPBS and fixed with 4% PFA for 20 min. Immunostaining of endomembrane $PI(4,5)P_2$ was performed as previously described (Hammond *et al*, 2009), except that PBS was used as buffer and 1% BSA for blocking. Briefly, cells were fixed, permeabilized with 20 µM digitonin for 5 min, blocked with 1% BSA, and stained with $PI(4,5)P_2$ antibody. Endomembrane pFAK antibody staining was performed in the same manner as endomembrane $PI(4,5)P_2$ staining.

## Western blotting

Cells were washed with ice-cold PBS and lysed in buffer containing HEPES pH 7.4, 1% NP40, 150 mM NaCl, 5 mM EDTA, 25 µg/ml chymostatin, 25 µg/ml leupeptin, 25 µg/ml antipain hydrochloride, 25 µg/ml pepstatin A, 10 mM sodium fluoride, 10 mM orthovanadate, and 1 mM PMSF. Equal amounts of protein were loaded onto Mini-Protein TGX Stain-Free gels and transferred onto PVDF membrane. Membranes were blocked with 1% BSA or 5% skim milk in TBS containing 0.1% Tween-20 for 30 min, and subsequently probed with primary antibodies at 4°C overnight. After washing with TBS containing 0.1% Tween-20, membranes were incubated with secondary antibodies at RT for 1 h. Membrane were washed, incubated with ECL Clarity (Bio-Rad, Cat#170-5061), and imaged with a ChemiDoc MD Imaging System (Bio-Rad). Band intensities were analyzed by using ImageJ software and normalized to total protein content quantified with the Stain-Free technology (Bio-Rad).

## Generation of endo-mEGFP-ORP2 and Degron-mEGFP-ORP2 cells

N-terminal miniIAA7-mEGFP tagging of endogenous ORP2 loci was conducted with CRISPR-Cas9-mediated homology-directed repair (HDR) as previously described (Li *et al*, 2019). For constructing donor vector, homology arms were amplified from A431 genomic DNA using primers: CTCACTGCGATCTCTGCCTC and GGGTTTCAGGCTCTTCCCTG for LH-RH, to be used as templates for LH and RH separately, with primers CTCACTGCGATCTCTGCCTC and AGCCCTCCATCCTTCAGCAGCCAGCCTCC for LH, and GGGTTTCAGGCTCTTCCCTG and AGGCGGATCCAACGGAGAGGAAGAATTCTTTGATGCCG for RH. MiniIAA7-mEGFP tag was amplified from pGL3-miniIAA7-mEGFP-LMNA (Li *et al*, 2019) using primers: (CTGCTGAAGGATGGAGGGCTTCTCTGAGACCG and CCTCTCCGTTGGATCCGCCTCCGCCAGAT). Overlap PCR was performed to assemble PCR fragments (LH, MiniIAA7-mEGFP and RH as templates) using primers: ATGCACGCGTGGGACTATAGGCGTGCGCC and ATGCGTCGACTCACATGCCCATGAGCCCTG. The PCR fragment was then cloned into pGL3 vector using SalI and MluI restriction sites to obtain pGL3-miniIAA7-mEGFP-ORP2 vector. SgRNAs (sense: CACCGCTGCTGAAGGATGAACGGAG, antisense: AAACCTCCGTTCATCCTTCAGCAGC) were synthesized as two unphosphorylated primers, annealed and inserted into BbsI-cut pCas9-sgRNA vector. Cells were transfected with the pGL3-miniIAA7-mEGFP-ORP2 and pCas9/sgRNA and selected with 1 µg/ml puromycin for 2 days to eliminate untransfected cells and high mEGFP-ORP2 expressing cells were enriched from the HDR pool by FACS and single clones were selected by limited dilution. To generate Degron-mEGFP-ORP2 cells, the homozygously tagged single clone was transfected with pCas9-gAAVS1 and pSH-EFIRES-P-AtAFB2 and selected with 1 µg/ml puromycin for 6 days. To sequence the modified genome region, genomic DNA was extracted by NucleoSpin Tissue (Macherey-Nagel, 740952), amplified by PCR using primers (forward: TTCTGGAAGCTAAGTATGC, reverse: CACACACTGCTTTACTGAG), purified and DNA sequencing was performed using the same primers (Appendix Fig S2). MiniIAA7-mEGFP-ORP2 cells, which do not express AtAFB2, were used as a control. For ORP2 depletion, degron-ORP2 cells were treated with 100 µg/ml indole-3-acetic acid (IAA) for the indicated time.

## Quantitative reverse transcription PCR

Total RNAs were isolated using the NucleoSpin RNA isolation kit (Macherey-Nagel, 740955-250), and 1 µg of total RNA was transcribed using SuperScript VILO cDNA synthesis kit (Invitrogen). Quantitative reverse transcription PCR was performed using Light Cycler 480 SYBR Green I Master Mix (Roche) and a Light Cycler 480 II (Roche). Reaction steps: 95°C for 15 min and 40 cycles of 95°C 15 s, 60°C 30 s, and 72°C 10 s. Relative quantities of mRNAs were normalized to 18S. Primers for 18S: GTCTTGTAGTTGCCGTCGTCCTTG (forward) and GTCTTGTAGTTGCCGTCGTCCTTG (reverse), for 3-Hydroxy-3-Methylglutaryl-CoA Reductase (HMGCR): ATGGAAACTCATGAGCGTGGT (forward) and AGCTCCCATCACCAAGGAGT (reverse), and for LDL receptor: TGGCTGCGTTAATGTGACACTC (forward) and AGCCGATCTTAAGGTCATTGC (reverse).

## [³H]oleic acid incorporation into cholesteryl esters

For analyzing LDL-induced cholesterol esterification, degron-ORP2 and control cells were cultured in 5% LPDS for 24 h, washed with PBS, and treated with 50 µg/ml LDL for 0–6 h. Cells were then supplemented for 2 h with [³H]oleic acid (5 µCi/ml, Perkin Elmer) in 2% defatted BSA prepared in 5% LPDS medium. During LDL and

oleic acid loading, the degron-ORP2 cells were supplemented with 100 μg/ml IAA as indicated. The cells were collected by scraping in cold 2% NaCl and lipids were extracted as described (Bligh & Dyer, 1959). Dried lipids were dissolved in chloroform/methanol (9:1), resolved by TLC using hexane/diethyl ether/acetic acid (80:20:1) as the mobile phase and dried plates were stained with iodine vapor. The cholesteryl ester bands were scraped and radioactivity measured by liquid scintillation counting. The results were corrected for the volume and procedural losses and plotted against the total amount of protein in the sample.

### Transferrin uptake

Cells were seeded in 24-well plates (75,000 cells per well) with four replicates for each time point/treatment. A separate identical plate was prepared for quantification of surface transferrin receptor. On the following day, cells were serum-starved at 37°C with pre-warmed serum-free media (containing or not the respective chemical treatments for each condition) for 1 h before the experiment. Methyl-β-cyclodextrin was used at 5 mM for 1 h. For uptake experiments, 10 μg/ml Alexa Fluor-647-labeled human transferrin (Thermo Fisher Scientific, T23366) were added to cells at 37°C for 8 min. For the quantification of surface transferrin receptor, cells were placed on ice and incubated with Alexa Fluor-647-labeled human transferrin (10 μg/ml) for 1 h. After incubation, cells were washed with PBS, detached with trypsin (for uptake experiments) or Accutase (Sigma, A6964) (for surface receptor measurement) followed by addition of ice-cold serum-containing media. Cells were transferred to 1.5-ml tubes, spun, washed with ice-cold PBS, and fixed with 4% PFA for 10 min at 4°C. Fixed cells were spun to remove fixative, resuspended in PBS, and transferred to FACS tubes. Fluorescence was measured using a BD LSR Fortessa flow cytometer (BD Biosciences) using appropriate filters. For each condition, 5,000 cells were measured. The median value for each replicate was used as a single experimental point. Normalized transferrin uptake was calculated by dividing the fluorescence signal of the uptake experiment for each condition by the average fluorescence signal of the surface receptor experiments for the same condition.

### Cell fractionation

Cells were washed with PBS on ice, scraped in PBS, and centrifuged at 2,000 rpm (380 *g*) for 10 min at 4°C. The cell pellets were resuspended in PBS containing 10 mM sodium fluoride, 10 mM orthovanadate, 1 mM PMSF, 25 μg/ml chymostatin, 25 μg/ml leupeptin, 25 μg/ml antipain hydrochloride, and 25 μg/ml pepstatin A, passed through a 25-gage needle for 100 times and centrifuged at 6,000 rpm (3,420 *g*) for 10 min at 4°C. The supernatant (post nuclear fraction) was centrifuged in a TLA-100 rotor at 50,000 rpm (109,000 *g*) for 1 h at 4°C. The supernatant (cytosol) and the pellet (membrane) were resuspended in PBS containing above inhibitors and SDS sample buffer. Both cytosol and membrane samples were boiled for 10 min and similar volumes loaded for Western blotting.

### Protein binding to liposomes and GUVs

Lipid compositions are indicated in the figure legends. A 1 mM lipid solution was mixed, dried under a stream of nitrogen gas and dried

further by Speedvac for 1 h, hydrated in HEPES (pH 7.4) with 100 mM NaCl, and vortexed for 1 h. Liposome co-sedimentation was performed with concentration of 50 μM lipids and 1 μM purified FAK FERM protein. The protein was pre-cleared by centrifugation at 100,000 rpm (436,000 *g*) for 30 min at 4°C with an Optima MAX Ultracentrifuge equipped with TLA-100 rotor (Beckman Coulter, Inc) to remove aggregates. Then, liposomes and protein were mixed and incubated at RT for 30 min, the mixture was centrifuged at 100,000 rpm (436,000 *g*) for 30 min. The pellet was collected as the bound fraction and supernatant as the non-bound fraction. For measuring protein binding to GUVs, a 5% (w/w) solution of polyvinyl alcohol (PVA, MW 145,000, Merck KGaA) was prepared by stirring PVA in water while heating at 90°C. PVA-coated substrates were prepared by spreading 20 μl of PVA solution on a microscope coverslip (18 mm in diameter), which was then dried for 30 min at 50°C. 10 μl of lipids dissolved in chloroform (1 mM) were spread on the dried PVA film and placed under vacuum for 30 min to evaporate the solvent. The lipid film was allowed to swell for 1 h in 500 μl of 5 mM HEPES (pH 7.4), 100 mM NaCl, and 20 mM sucrose. The formed GUVs were collected and incubated with 1 μM Alexa 488-labeled FAK FERM protein for 30 min. The coverslips were coated with β-casein (Sigma) to avoid non-specific protein binding. Images were acquired with Nikon Eclipse Ti-E microscope, x100 PlanApo VC oil objective.

### Image quantification

Quantifications were performed with ImageJ FIJI. Background-subtracted images were quantified as mean intensity or using Mander's overlap coefficient as a measure of colocalization.

### Statistical analysis

Statistical significance was determined by Non-parametric Kruskal–Wallis analysis of variance with Bonferroni error correction, two-tailed and paired (where indicated) Student's *t*-test, or one-way ANOVA with Tukey's post hoc test. The number of replicates (*n*) used for calculating statistics is specified in the Figure legends. Differences were considered statistically significant at $P < 0.05$. Results are expressed as mean ± SD or ± SEM of the indicated number of observations.

## Data availability

This study includes no data deposited in external repositories.

**Expanded View** for this article is available online.

### Acknowledgments

We thank Anna Uro for excellent technical assistance, Shiqian Li for help in establishing degron-ORP2 cells, Yosuke Senju for help in liposome assays, Young Ah Kim for BODIPY-cholesteryl linoleate, Mikko Liljeström for help with image analysis, and Biocenter Finland and HiLIFE for infrastructure support (BioImaging platform, Flow cytometry unit). This study was supported by the Academy of Finland (grants 307415 and 324929 to E.I., 322647 to V.M.O.), Sigrid Juselius Foundation (E.I., V.M.O., L. A.-S.), Fondation Leducq (grant 19CVD04) (E.I.), Jane and Aatos Erkko Foundation (E.I.), Magnus Ehrnrooth Foundation

(K.T., V.M.O.), and the Finnish Foundation for Cardiovascular Research (V.M.O.). L.-A.S. acknowledges support from Helsinki Institute of Life Science (HiLIFE, start-up grant) and D.L. from the Ministry of Science, Innovation and Universities for the Spanish State Research Agency Retos Grant RTI2018-099318-B-I00, cofounded by the European Regional Development Fund (FEDER).

## Author contributions
Study design: EI and KT; Experiments and data analysis: KT, KK, LV, and LA-S; Tools and techniques: DL and VMO; Study supervision: DL, VMO, and EI; Manuscript writing: KT and EI.

## Conflict of interest
The authors declare that they have no conflict of interest.

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
