## [Review Process File · The EMBO Journal]

ORP2 couples LDL-cholesterol transport to FAK activation by endosomal cholesterol/PI(4,5)P₂ exchange

Kohta Takahashi, Kristiina Kanerva, Lauri Vanharanta, Leonardo Almeida-Souza, Daniel Lietha, Vesa Olkkonen, and Elina Ikonen

DOI: [10.15252/embj.2020106871](https://doi.org/10.15252/embj.2020106871)

Corresponding author(s): *Elina Ikonen (elina.ikonen@helsinki.fi)*

Review Timeline:

Submission Date:	23rd Sep 20
Editorial Decision:	21st Oct 20
Revision Received:	23rd Jan 21
Editorial Decision:	23rd Feb 21
Revision Received:	29th Mar 21
Editorial Decision:	25th Apr 21
Revision Received:	30th Apr 21
Accepted:	3rd May 21

Editor: Elisabetta Argenzio

Transaction Report:

Thank you for submitting your manuscript entitled "ORP2 couples LDL-cholesterol transport to FAK activation by endosomal cholesterol/PI(4,5)P2 exchange" (EMBOJ-2020-106871) to The EMBO Journal. Your study has been sent to three referees for evaluation and we have now received their reports, which are enclosed below for your information.

As you can see, while the referees find your work potentially interesting, they also raise several major issues that need to be addressed before they can support publication in The EMBO Journal. In particular, referee #1 requests you to strengthen the proposed mechanism of LDL transfer between late and recycling endosomes via transient contacts and to better investigate the function of ORP2 in your model. Reviewer #2 finds that the work on integrin adhesion site dynamics and FAK activation is underdeveloped and that the connection between ORP2 and FAK is unclear. Referee #3 asks you to place ORP2-dependent cholesterol transfer in the context of overall LDL cholesterol mobilization from LE/Lys and to address the importance of this route as compared with the other ones.

We agree with the referees that these are important points and addressing these and all other issues will be essential to pursue publication of this study in The EMBO Journal. Please note that strong support from the referees would also be needed for publication here. Given the overall interest of your study, I would like to invite you to submit a new version of the manuscript revised according to the referees' requests. I should add that it is The EMBO Journal policy to allow only a single round of revision, and acceptance of your manuscript will therefore depend on the completeness of your responses in the revised version.

We generally grant three months as standard revision time. As we are aware that many laboratories cannot function at full capacity owing to the COVID-19 pandemic, we may relax this deadline. Also, we have decided to apply our 'scooping protection policy' to the time span required for you to fully revise your manuscript and address the experimental issues highlighted herein. Nevertheless, please inform us as soon as a paper with related content is published elsewhere.

When preparing your letter of response to the referees' comments, please bear in mind that this will form part of the Review Process File and will therefore be available online to the community. For more details on our Transparent Editorial Process, please visit our website:
http://emboj.embopress.org/about#Transparent_Process

Before submitting your revised manuscript, deposit any primary datasets and computer code produced in this study in an appropriate public database (see <http://msb.embopress.org/authorguide#dataavailability>). Please remember to provide a reviewer password, in case such datasets are not yet public. The accession numbers and database names should be listed in a formal "Data Availability" section (placed after Materials & Method). Provide a "Data availability" section even if there are no primary datasets produced in the study.

Feel free to contact me if you have any questions about the submission of the revised manuscript to The EMBO Journal. I thank you again for the opportunity to consider this work for publication and look forward to your revision.

Referee #1:

Takahashi et al. show evidence that ORP2/OSBPL2, a cytoplasmic protein previously implicated in exchange of cholesterol for PI(4,5)P2 at the plasma membrane, mediates delivery of LDL-derived cholesterol from late endosomes to recycling endosomes. The authors first observed that LDL enhances both focal adhesion assembly and disassembly in a FAK-dependent manner, and that LDL-cholesterol is delivered to FAK- and integrin-containing endosomes. Moreover, they found evidence that FAK regulates cholesterol export from NPC1-positive endosomes. The authors then employed an elegant auxin-inducible degron system to achieve acute depletion of ORP2 expressed at its endogenous locus. This led to a reduction in LDL-derived cholesterol at the plasma membrane and reduced cell adhesion. Interestingly, the authors also observed that acute depletion of ORP2 impaired transfer of cholesterol from late endosomes to recycling endosomes positive for integrin $\beta 1$. ORP2 depletion was accompanied by the failure of endosomes to gain PI(4,5)P2 upon LDL loading. Finally, evidence is presented that cholesterol facilitates the association of FAK with PI(4,5)P2-containing membranes. The authors conclude that ORP2 controls FAK activation and delivery of LDL-cholesterol to the plasma membrane by promoting bidirectional cholesterol/PI(4,5)P2 exchange between late endosomes and recycling endosomes.

Given the importance of cellular cholesterol uptake, the take-home message of this manuscript is very interesting. In general the conclusions are well supported by the data, which have been obtained through use of cutting-edge technology such as inducible gene knockout and advanced microscopy. Nevertheless, some points should be addressed prior to publication.

1. The antibody staining of PI(4,5)P2 in this manuscript shows PI(4,5)P2 mainly localized to numerous cytoplasmic puncta, with only little PI(4,5)P2 detected on the plasma membrane. This is in contrast to the view that the bulk of PI(4,5)P2 is localized to the plasma membrane (Di Paolo and De Camilli, *Nature*, 2006). The authors should verify their PI(4,5)P2 labeling with an alternative marker for this phosphoinositide, such as a PI(4,5)P2-specific PH domain.

2. The proposed mechanism of LDL transfer between late endosomes and recycling endosomes via transient contacts is a bit surprising in light of current literature, which has mainly considered vesicular trafficking between early/late endosomes and recycling endosomes. The model is mainly based on live imaging, which makes a fairly compelling case. However, it is assumed that $\beta 1$ -integrin is a marker of recycling endosomes, which is not necessarily the case. This integrin subunit can also be found in late endosomes, at least under conditions when fibronectin is internalized (Lobert et al., *Dev.Cell*, 2010). In the movie there is also no marker of late endosomes other than BODIPY-cholesterol, which could also label other compartments. Given the originality of this finding, I think the authors need to strengthen their model with additional evidence based on alternative markers.

3. If the model of contact-mediated cholesterol transfer is correct, it is a bit difficult to understand the exact function of ORP2. In the model (Fig. 8) ORP2 is indicated as a cytosolic shuttle of cholesterol, but why do we then need kiss-and-run contacts for cholesterol transfer? An alternative possibility is that ORP2 could be required for formation of short-lived contact sites, and the authors should investigate whether depletion of ORP2 prevents kiss-and-run contacts between late and recycling endosomes. The authors might consider the model for contact site-mediated cholesterol transfer from the ER to the TGN, mediated by the cholesterol- and PI(4)P-binding protein OSBP

(Mesmin et al., Cell, 2013).

Referee #2:

Summary: This study analyzes A431 cells plated on fibronectin for alterations in vesicular trafficking upon starvation and stimulation with LDL. Results are from analyses of widefield or confocal microscopic images of marker-tagged vesicles using fixation and indirect antibody staining or live cell imaging with fluorescent proteins. The cell model focuses on LDL-derived cholesterol transfer from late endosomes through recycling-endosomes to plasma membrane (PM). Results are shown that there is a connection between LDL addition, cholesterol accumulation at the plasma membrane, cell adhesion site (integrin-matrix) remodeling as measured by assembly-disassembly rates, and focal adhesion kinase (FAK) tyrosine phosphorylation. A new auxin-tagged protein degradation cell model was used to explore the role of ORP2 loss in endosome dynamics.

Reviewer Opinion: This is a potentially interesting study presenting new ideas about LDL, cholesterol transport, and vesicle trafficking in cells. However, the connections and conclusion made with regard to integrin adhesion site dynamics and FAK activation are underdeveloped and results from the use of a pharmacological FAK inhibitor were many times over-interpreted. There is also concern with the timing of observed changes in adhesions (early, minutes) and FAK activation (late, hours) after LDL treatment. In contrast, purified cholesterol could trigger maximal FAK activation in 2 minutes (what does this mean?). Better controls are needed for proof of anti-integrin and anti-FAK pY397 staining as the several of the images presented look mostly like non-specific staining. The connection between ORP2 and FAK remains unclear, although both of these proteins can bind common intermediaries. It would be helpful to have biochemical or cell fractionation analyses to support the most important conclusion regarding changes in vesicle trafficking.

Specific Points:

1. Methods: The use of widefield microscopy for trafficking studies is unclear. Confocal methods need expansion. How were cells chosen? How many regions of interest per cell? The images acquired are also different in means of confluency and where selected analysis point is located.
2. The n between experiments varies a lot and should be better explained. Related to this is the selection of statistics and whether SEM should be replaced with SD. That in case all the single cells analyzed are considered for n. If two separate experiments are used to calculate the mean of experiment and based on these SEM is calculated, then at least 3 experiments are needed.
3. Why are the A431 cells plated onto fibronectin? Why studies with just A431 cells?
4. The level of PF228 used is very high (10 μ M). In most cells, the IC50 is below 1 micromolar. The authors need to perform a dose- and time- experiment to determine optimal concentrations for PF228 and A431 cells.
5. GFP-talin is not standard for analysis of adhesion dynamics. Talin is known to regulate contractile-based changes in mature adhesions. Additionally, the size of the fusion protein (270 kDa) and protein folding issues can cause artifacts. The online method tool was developed for cells transfected with GFP-paxillin.

6. FAK can promote adhesion turnover. Less is known about the role of FAK in adhesion formation (knockouts show that it is not essential). FAK inhibitor treatment blocks the "cycle" of adhesion turnover/formation. Thus, conclusions associated with just inhibitor use need to be tempered.

7. Fig. 1F. Why is FAK activation "late" [1 h] when adhesion changes have finished by 45 minutes? The timing is not clear. EV Figure 1. Why is maximal FAK activation by cholesterol at 2 min? in A431 cells, FAK clusters with cholesterol and caveolin at the plasma membrane (Park et al. J. Pathol. 218:337, 2009).

8. Need proof of specificity of anti-integrin and anti-pY397 FAK staining. Since the authors have the GFP-FAK expression vector, this could be used in experiments.

9. If the Degron-mEGFP-ORP2 cells (clones) were created for this study they should be sequenced at least from the region where modifications were made. The ORP2 KO could be showed also by WB method and that it does not have adverse effects.

10. Finally the results in Figure 1 E and in Expanded View Figure 1 B create confusion as the assembly rate with FAK WT overexpression -LDL is bigger compared to +LDL. When FAK is overexpressed the LDL effect on disassembly is very small whereas in Figure 1 E it again seems to have a much larger effect. The issue is reproducibility.

11. The liposome data in Figure 7 does not match the flow of the paper. Previous studies have shown that lipid binding is not sufficient to trigger FAK catalytic activation (as opposed to conformational changes).

12. The loss of ORP2 protein on total FAK tyrosine phosphorylation (EV Fig. 6) is not impressive (and the blots may be from different membranes). Author need to show that FAK pY307 changes are happening on vesicles and maybe not at mature adhesions.

Referee #3:

EMBOJ-2020-106871

Takahashi et al.

This manuscript examines the trafficking of LDL cholesterol from lysosomes to recycling FAK/integrin-containing endosomes. They show that ORP2 mediates transfer of LDL cholesterol to these organelles, the delivery of which activates FAK activation and generation of PI(4,5)P₂. ORP2 expression also increases PI(4,5)P₂ in NPC1-containing LE/Ly. A model in which ORP2 delivers LDL cholesterol to recycling endosomes in exchange for PI(4,5)P₂ and retrograde delivery to LE/Ly is proposed.

The manuscript addresses an important question in intracellular cholesterol trafficking, namely the mechanism(s) involved post lysosomal movement of cholesterol. The present study focuses on the role of ORP2 in delivery of cholesterol to beta1-integrin/FAK recycling endosomes. The data presented is of high quality and generally support the conclusions. A general weakness of the study is perhaps that it is too focused. The manuscript would be strengthened by placing ORP2-dependent cholesterol transfer in the context of overall mobilization of LDL cholesterol from LE/Lys and addressing the relative importance of this route. There is also a concern with respect to the

heavy reliance on the D4H probe. These concerns notwithstanding, this study could be a solid contribution to the field.

Specific points:

1. LDL cholesterol has been reported to distribute from LE/Ly to multiple organelles, including PM, ER, Golgi, mitochondria and peroxisomes. Direct movement to recycling endosomes is not well described, as recycling endosomes have been generally shown to receive internalized cholesterol from the PM. What is the evidence that the cholesterol delivered by ORP2 has not first been trafficked to the PM and then internalized to recycling endosomes? (see also concern raised in point #2)
2. The authors readily acknowledge the limitations of the fluorescent cholesterol probe used in this study, TopFluor Chol, and therefore in Fig 2 and for the remainder of the manuscript use intracellular expression of D4H to monitor cholesterol trafficking. While D4H is an attractive probe, it is less sensitive than a fluorescent cholesterol analog because it requires cholesterol mol% to be above a threshold of membrane enrichment to fluoresce. This creates a time lag that is readily apparent when comparing data in panels 2E/G and 2A. This could be addressed using more faithful analogs such as cholesterol alkynes. This is important to reconcile because reliance solely on the D4H probe obscures whether the LDL cholesterol is trafficking directly to the recycling endosomes or being re-internalized from the PM. There is certainly ample PI(4,5)P2 at the PM for ORP2 to pick up in exchange for cholesterol.
3. The study is highly focused on ORP2 cholesterol trafficking from LE/Ly to recycling endosomes and does not provide a sufficient context for the broader audience to understand the relative contribution of ORP2 to post-lysosomal cholesterol movement. Does ORP2 KD affect delivery of cholesterol to organelles other than recycling endosomes? For example, what is the effect of ORP2 LOF in delivery of cholesterol to the ER for re-esterification or for suppression of SREBP2 target gene expression, or in delivery of cholesterol to the PM?
4. General concern: It is unclear from reading methods and figure legends whether specific IF panels are epifluorescence imaging or confocal slices. This is important in interpreting IF as to whether there is true overlap rather than superimposition.

EMBOJ-2020-106871

Takahashi et al. ORP2 couples LDL-cholesterol transport to FAK activation by endosomal cholesterol/PI(4,5)P₂ exchange

Point-by-point responses to the Reviewers' queries

The changes made accordingly are indicated with blue font in the revised manuscript.

Referee #1:

Takahashi et al. show evidence that ORP2/OSBPL2, a cytoplasmic protein previously implicated in exchange of cholesterol for PI(4,5)P₂ at the plasma membrane, mediates delivery of LDL-derived cholesterol from late endosomes to recycling endosomes. The authors first observed that LDL enhances both focal adhesion assembly and disassembly in a FAK-dependent manner, and that LDL-cholesterol is delivered to FAK- and integrin-containing endosomes. Moreover, they found evidence that FAK regulates cholesterol export from NPC1-positive endosomes. The authors then employed an elegant auxin-inducible degron system to achieve acute depletion of ORP2 expressed at its endogenous locus. This led to a reduction in LDL-derived cholesterol at the plasma membrane and reduced cell adhesion. Interestingly, the authors also observed that acute depletion of ORP2 impaired transfer of cholesterol from late endosomes to recycling endosomes positive for integrin β1. ORP2 depletion was accompanied by the failure of endosomes to gain PI(4,5)P₂ upon LDL loading. Finally, evidence is presented that cholesterol facilitates the association of FAK with PI(4,5)P₂-containing membranes. The authors conclude that ORP2 controls FAK activation and delivery of LDL-cholesterol to the plasma membrane by promoting bidirectional cholesterol/PI(4,5)P₂ exchange between late endosomes and recycling endosomes.

Given the importance of cellular cholesterol uptake, the take-home message of this manuscript is very interesting. In general the conclusions are well supported by the data, which have been obtained through use of cutting-edge technology such as inducible gene knockout and advanced microscopy. Nevertheless, some points should be addressed prior to publication.

We thank the Reviewer for the highly encouraging and positive remarks on our study. We have now addressed all the specific points raised, as detailed below. We wish to thank the Reviewer for the insightful suggestions for additional experiments that helped to improve this manuscript.

1. The antibody staining of PI(4,5)P₂ in this manuscript shows PI(4,5)P₂ mainly localized to numerous cytoplasmic puncta, with only little PI(4,5)P₂ detected on the plasma membrane. This is in contrast to the view that the bulk of PI(4,5)P₂ is localized to the plasma membrane (Di Paolo and De Camilli, *Nature*, 2006). The authors should verify their PI(4,5)P₂ labeling with an alternative marker for this phosphoinositide, such as a PI(4,5)P₂-specific PH domain.

Thank you for the remark. We used an PI(4,5)P₂ antibody staining procedure specifically optimized for endosomal PI(4,5)P₂ detection by G. Schiavo and coworkers (Hammond *et al*, 2009), and this explains why the antibody detects cytoplasmic puncta. The same antibody can also be used for detecting plasma membrane PI(4,5)P₂ (with another staining procedure described in the same paper; see Figure for Reviewer below). However, this staining protocol detects A431 plasma membrane at cell-cell junctions poorly (as is obvious in the Figure) and is not compatible with detection of endogenous GFP-ORP2 (the signal of which becomes too weak), and therefore this protocol could not be employed.

As requested, we have performed additional experiments using a PI(4,5)P₂ specific PH domain probe. We employed the PLCdelta-PH domain to investigate changes in plasma membrane PI(4,5)P₂ upon FAK inhibition or acute ORP2 degradation. We show that in PF228 treated cells, where the PI(4,5)P₂ antibody detects a reduced endomembrane PI(4,5)P₂ signal, the PLCdelta PH-domain probe detects a reduced PM signal (Fig EV3D,E). This PH-domain has been previously used by us and others to demonstrate an increase in PI(4,5)P₂ upon ORP2 depletion (upon constitutive ORP2 knockout or upon 3 days ORP2 silencing) (Koponen *et al*, 2020; Wang *et al*, 2019). Interestingly, by using this probe we observed no difference in plasma membrane PI(4,5)P₂ immediately after ORP2 removal (up to 4 h, when the effects on LDL-cholesterol delivery are already evident), but only at longer times (3 days of ORP2 degradation; in consistence with Koponen *et al.*, 2020). These data are included in Fig EV4E-G. In addition, we analyzed whether a canonical plasma membrane PI(4,5)P₂ dependent membrane trafficking process (clathrin-mediated endocytosis of transferrin) was affected and found it not to be changed within the first 4 h of ORP2 removal. This is shown in Fig EV4H. Together, these data suggest that a change in PM PI(4,5)P₂ develops gradually upon ORP2 removal.

2. The proposed mechanism of LDL transfer between late endosomes and recycling endosomes via transient contacts is a bit surprising in light of current literature, which has mainly considered vesicular trafficking between early/late endosomes and recycling endosomes. The model is mainly based on live imaging, which makes a fairly compelling case. However, it is assumed that b1-integrin is a marker of recycling endosomes, which is not necessarily the case. This integrin subunit can also be found in late endosomes, at least under conditions when fibronectin is internalized (Lobert *et al.*, *Dev.Cell*, 2010). In the movie there is also no marker of late endosomes other than BODIPY-cholesterol, which could also label other compartments. Given the originality of this finding, I think the authors need to strengthen their model with additional evidence based on alternative markers.

Following the Reviewer's suggestion, we have now included additional markers for the compartments of interest: dextran for late endosomal organelles and transferrin for recycling endosomes. We show that under the conditions used, BODIPY-cholesterol deriving from the hydrolysis of BODIPY-cholesterol esters incorporated into LDL particles, is first localized in dextran positive compartments (Fig EV2A), in agreement with our earlier findings (Kanerva *et al*, 2013). Furthermore, we show that BODIPY-cholesterol is reaching transferrin positive endosomes during the chase. These data are now included in Fig EV2C-E. We then also used dextran and transferrin in the experiments to address the transient contacts between late and recycling endosomes by rapid live cell imaging (to address point 3 below).

3. If the model of contact-mediated cholesterol transfer is correct, it is a bit difficult to understand the exact function of ORP2. In the model (Fig. 8) ORP2 is indicated as a cytosolic shuttle of cholesterol, but why do we then need kiss-and-run contacts for cholesterol transfer? An alternative possibility is that ORP2 could be required for formation of short-lived contact sites, and the authors should investigate whether depletion of ORP2 prevents kiss-and-run contacts between late and recycling endosomes. The authors might consider the model for contact site-mediated cholesterol transfer from the ER to the TGN, mediated by the cholesterol- and PI(4)P-binding protein OSBP (Mesmin *et al.*, *Cell*, 2013).

This is a good – and challenging – question and remains to be comprehensively addressed in future studies. However, as proposed by the Reviewer, we have now investigated if ORP2 affects kiss-and-run contacts between late and recycling endosomes, by using dextran and transferrin as well-established tracers for these compartments. We found that acute ORP2 removal did have a major effect on such contacts, by making them less prevalent but more persistent. We have included these data in Fig EV5A-C. While we do not fully understand this observation mechanistically at this point, the effect is clear and reveals that ORP2 removal rapidly impacts on dynamic contacts between late and recycling endosomes and that ORP2 is important for maintaining the transient nature of membrane contacts between these organelles.

Referee #2:

Summary: This study analyzes A431 cells plated on fibronectin for alterations in vesicular trafficking upon starvation and stimulation with LDL. Results are from analyses of widefield or confocal microscopic images of marker-tagged vesicles using fixation and indirect antibody staining or live cell imaging with fluorescent proteins. The cell model focuses on LDL-derived cholesterol transfer from late endosomes through recycling-endosomes to plasma membrane (PM). Results are shown that there is a connection between LDL addition, cholesterol accumulation at the plasma membrane, cell adhesion site (integrin-matrix) remodeling as measured by assembly-disassembly rates, and focal adhesion kinase (FAK) tyrosine phosphorylation. A new auxin-tagged protein degradation cell model was used to explore the role of ORP2 loss in endosome dynamics.

Reviewer Opinion: This is a potentially interesting study presenting new ideas about LDL, cholesterol transport, and vesicle trafficking in cells. However, the connections and conclusion made with regard to integrin adhesion site dynamics and FAK activation are underdeveloped and results from the use of a pharmacological FAK inhibitor were many times over-interpreted. There is also concern with the timing of observed changes in adhesions (early, minutes) and FAK activation (late, hours) after LDL treatment. In contrast, purified cholesterol could trigger maximal FAK activation in 2 minutes (what does this mean?). Better controls are needed for proof of anti-integrin and anti-FAK pY397 staining as the several of the images presented look mostly like non-specific staining. The connection between ORP2 and FAK remains unclear, although both of these proteins can bind common intermediaries. It would be helpful to have biochemical or cell fractionation analyses to support the most important conclusion regarding changes in vesicle trafficking.

We thank the Reviewer for the expert evaluation of our manuscript and constructive suggestions on how to further improve it. We apologize that some of the arguments were not cautiously enough formulated and some controls were missing. Regarding the timing of cholesterol induced changes, the hydrolysis of LDL particles in the acidic endosomal milieu is gradual as is the release of cholesterol therefrom and hence, any downstream effects as well. When adding a supraphysiological dose of purified cholesterol from a cyclodextrin complex, cholesterol is incorporated within minutes to membranes and can cause downstream effects more rapidly and synchronously. Regarding the Reviewer's suggestion for cell fractionation analyses, we have now included membrane fractionation data demonstrating that FAK membrane association is affected by acute ORP2 removal. Of note, the most important conclusions in our study regarding changes in lipid trafficking are between late and recycling endosomes, and available cell fractionation schemes do not allow us to dissociate these two compartments sufficiently to pinpoint dynamic changes between their lipid compositions biochemically. Regarding the specific points raised, we have carefully scrutinized them and have revised the manuscript, as outlined in the point-by-point responses below.

Specific Points:

1. Methods: The use of widefield microscopy for trafficking studies is unclear. Confocal methods need expansion. How were cells chosen? How many regions of interest per cell? The images acquired are also different in means of confluency and where selected analysis point is located.

Thank you for raising these points. We have now expanded the Methods sections, to more explicitly explain the use of widefield microscopy and confocal microscopy, as well as cell selection of cells and cell areas for imaging, including the effect of confluency.

2. The n between experiments varies a lot and should be better explained. Related to this is the selection of statistics and whether SEM should be replaced with SD. That in case all the single cells analyzed are considered for n. If two separate experiments are used to calculate the mean of experiment and based on these SEM is calculated, then at least 3 experiments are needed.

Thank you for the remark. We have now more carefully explained the n between experiments and replaced SEM with SD, where needed. Minimally 2 independent experiments were included for all data points.

3. Why are the A431 cells plated onto fibronectin? Why studies with just A431 cells?

We chose A431 cells because we have used them extensively for studying lipid trafficking, including studies on LDL-cholesterol trafficking (e.g. (Heybrock *et al*, 2019; Kanerva *et al.*, 2013; Salo *et al*, 2019; Santinho *et al*, 2020) and we are most experienced in using the improved AID system in this cell model (Li *et al*, 2019). We used fibronectin coating to promote the use of integrin dependent adhesion mechanisms by the cells and have now added this notion.

4. The level of PF228 used is very high (10 μ M). In most cells, the IC50 is below 1 micromolar. The authors need to perform a dose- and time- experiment to determine optimal concentrations for PF228 and A431 cells.

Indeed, it is true that PF228 was originally reported to have an IC50 at a high nanomolar range (Slack-Davis *et al*, 2007). However, a large number of later studies referring to this work has used the compound at a 10 μ M concentration in various cell types, including recent publications (Kalappurakkal *et al*, 2019; Nader *et al*, 2016; Panagiotakopoulou *et al*, 2018). We performed dose- and time-experiments in A431 cells and found that also in these cells high concentrations are needed (Figure for Reviewer, below). Therefore, we used PF228 at 10 μ M concentration. We appreciate that high drug concentrations are more prone to cause unspecific effects and have therefore now also included data showing that the effect of PF228 on LDL-derived cholesterol accessibility both in endomembranes and in the PM can also be observed upon FAK silencing (Fig 3G, H and Fig EV3C).

5. GFP-talin is not standard for analysis of adhesion dynamics. Talin is known to regulate contractile-based changes in mature adhesions. Additionally, the size of the fusion protein (270 kDa) and protein folding issues can cause artifacts. The online method tool was developed for cells transfected with GFP-paxillin.

The reviewer is right. We have therefore generated a stable cell line expressing GFP-paxillin and reproduced the experiments on LDL-dependent adhesion dynamics in these. The results and conclusions are the same as those obtained with GFP-talin. The data are included in Fig 1D, E.

6. FAK can promote adhesion turnover. Less is known about the role of FAK in adhesion formation (knockouts show that it is not essential). FAK inhibitor treatment blocks the "cycle" of adhesion turnover/formation. Thus, conclusions associated with just inhibitor use need to be tempered.

We agree with the Reviewer and have now specifically mentioned that the FAK inhibitor blocks the cycle of adhesion turnover (p. 8). Furthermore, we have strengthened the data by showing that the effects of the FAK inhibitor on LDL-cholesterol endosomal and PM delivery were reproduced by FAK silencing (Fig 3G, H and Fig EV3C).

7. Fig. 1F. Why is FAK activation "late" [1 h] when adhesion changes have finished by 45 minutes? The timing is not clear. EV Figure 1. Why is maximal FAK activation by cholesterol at 2 min? in A431 cells, FAK clusters with cholesterol and caveolin at the plasma membrane (Park et al. J. Pathol. 218:337, 2009).

Thank you for raising these points. In cells, the activation of FAK by WB can be seen already at 30 min and is evident at 1 h upon LDL addition (Fig 1F, Fig EV6), while the changes in focal adhesion dynamics are observed from 1-2 h LDL incubation onwards but not at 0-1 h LDL (Fig. EV1E). Thus, it appears that LDL-induced FAK activation precedes increased dynamics of plasma membrane adhesion sites and may result from LDL-cholesterol dependent activation of FAK in endomembranes. When cholesterol is added to cells directly from a methyl-beta-cyclodextrin complex (Fig EV1F), cholesterol reaches the plasma membrane and endomembranes within minute(s), because it does not require receptor-mediated endocytosis as opposed to LDL. Thus, Fig EV1F shows that FAK is enhanced upon acutely increasing membrane cholesterol. We have added this notion to the manuscript (p. 6) to clarify this experiment.

8. Need proof of specificity of anti-integrin and anti-pY397 FAK staining. Since the authors have the GFP-FAK expression vector, this could be used in experiments.

We have added the requested controls, i.e. specificity of anti-integrin and anti-pY397 FAK stainings for the applications used in this study, i.e. immunofluorescence microscopy or Western blotting. These data are provided in Appendix Fig 1.

9. If the Degron-mEGFP-ORP2 cells (clones) were created for this study they should be sequenced at least from the region where modifications were made. The ORP2 KO could be showed also by WB method and that it does not have adverse effects.

The Degron-mEGFP-ORP2 cells were indeed generated for this study and we have added information on the modified sequence (Appendix Fig 2), as requested. The lack of ORP2 signal upon degradation is provided in the WB in Fig EV4C.

10. Finally the results in Figure 1 E and in Expanded View Figure 1 B create confusion as the assembly rate with FAK WT overexpression -LDL is bigger compared to +LDL. When FAK is overexpressed the LDL effect on disassembly is very small whereas in Figure 1 E it again seems to have a much larger effect. The issue is reproducibility.

There is probably some misunderstanding here. In current Fig EV1C (previous EV1B), the assembly rate with FAK WT overexpression - LDL is actually smaller compared to the +LDL situation. The effect of LDL addition on focal adhesion disassembly varies between experiments but nevertheless there is a significant and reproducible increase upon LDL both in the GFP-paxillin overexpressing cells (revised Fig 1E) and in the FAK overexpressing cells (Fig EV1C). We wish to note that a relatively large variance between measurements is also observed in the original description of the assay (Berginski *et al*, 2011).

11. The liposome data in Figure 7 does not match the flow of the paper. Previous studies have shown that lipid binding is not sufficient to trigger FAK catalytic activation (as opposed to conformational changes).

Thank you for the remark. To bridge the cell imaging -based results in Figure 7 better with the liposome-based results, we have now included biochemical evidence showing that ORP2 controls the membrane association of FAK in cells: acute ORP2 removal abrogates the membrane association of endogenous FAK in LDL-loaded cells. These data are provided in Fig 7C.

12. The loss of ORP2 protein on total FAK tyrosine phosphorylation (Fig EV6) is not impressive (and the blots may be from different membranes). Author need to show that FAK pY307 changes are happening on vesicles and maybe not at mature adhesions.

The Reviewer is correct, the blots are from different membranes, because stripping and reprobing of filters results in substantially compromised specific recognition with these antibodies. However, this does not affect the conclusions. For clarity, we have added “loading control” lanes from the same blots in Fig EV6A. We have also included additional evidence to demonstrate that the FAK pY397 (we presume this is what the Reviewer means) signal increase is clearly punctate/vesicular, localizes juxtenuclearly and is thus morphologically distinct from labeling in mature adhesions (Fig EV6B).

Referee #3:

This manuscript examines the trafficking of LDL cholesterol from lysosomes to recycling FAK/integrin-containing endosomes. They show that ORP2 mediates transfer of LDL cholesterol to these organelles, the delivery of which activates FAK activation and generation of PI(4,5)P2. ORP2 expression also increases

PI(4,5)P₂ in NPC1-containing LE/Ly. A model in which ORP2 delivers LDL cholesterol to recycling endosomes in exchange for PI(4,5)P₂ and retrograde delivery to LE/Ly is proposed.

The manuscript addresses an important question in intracellular cholesterol trafficking, namely the mechanism(s) involved post lysosomal movement of cholesterol. The present study focuses on the role of ORP2 in delivery of cholesterol to beta1-integrin/FAK recycling endosomes. The data presented is of high quality and generally support the conclusions. A general weakness of the study is perhaps that it is too focused. The manuscript would be strengthened by placing ORP2-dependent cholesterol transfer in the context of overall mobilization of LDL cholesterol from LE/Lys and addressing the relative importance of this route. There is also a concern with respect to the heavy reliance on the D4H probe. These concerns notwithstanding, this study could be a solid contribution the field.

We thank the Reviewer for the highly positive assessment of our work and for the constructive and insightful suggestions on how to further improve it. While revising the manuscript, we have payed special attention to placing ORP2-dependent cholesterol transfer better in the context of the overall mobilization of LDL-cholesterol from late endosomal compartments, and also to using additional readouts besides the D4H probe. Please see our responses to the specific points raised below.

Specific points:

1. LDL cholesterol has been reported to distribute from LE/Ly to multiple organelles, including PM, ER, Golgi, mitochondria and peroxisomes. Direct movement to recycling endosomes is not well described, as recycling endosomes have been generally shown to receive internalized cholesterol from the PM. What is the evidence that the cholesterol delivered by ORP2 has not first been trafficked to the PM and then internalized to recycling endosomes? (see also concern raised in point #2)

This is a good question. The evidence for this is that we observe accessible LDL-derived cholesterol (using D4H) and fluorescent LDL-derived sterol (using TopFluor a.k.a. BODIPY-cholesterol) first in punctate intracellular structures that colocalize with late or recycling endosomal markers, before the pronounced increase in PM D4H or BODIPY-cholesterol labeling develops. However, as cholesterol reaching the PM is likely to diffuse in the plane of the membrane and not to stay concentrated as in endosomes, we cannot rule out the possibility that some cholesterol is delivered by ORP2 to the PM in parallel or even before it is delivered to recycling endosomes (see also our response to Point 2).

Considering that ORP2 acts as a cholesterol-PI(4,5)P₂ exchanger, we have now also assessed whether there are indications for rapid changes in PM PI(4,5)P₂ in acutely ORP2 depleted cells. Using the PLCdelta PH domain as a probe, we did not detect changes in PM PI(4,5)P₂ content immediately upon ORP2 removal (in 0-4 h, where endosomal cholesterol sequestration is observed) – rather only after several days of ORP2 degradation (Fig EV4E-G). The latter finding agrees with observations reported by us and others (Koponen *et al.*, 2020; Wang *et al.*, 2019). Moreover, we have now studied whether PI(4,5)P₂ dependent membrane trafficking at the level of the PM is perturbed upon acute ORP2 removal. We found clathrin-mediated endocytosis of transferrin not to be affected immediately upon acute (up to 4 h) ORP2 removal (Fig EV4H). Together, these data strengthen the idea that the increase in PM PI(4,5)P₂ content develops gradually and that the primary ORP2 cholesterol-PI(4,5)P₂ exchange function is at the level of endosomes.

2. The authors readily acknowledge the limitations of the fluorescent cholesterol probe used in this study, TopFluor Chol, and therefore in Fig 2 and for the remainder of the manuscript use intracellular expression of D4H to monitor cholesterol trafficking. While D4H is an attractive probe, it is less sensitive than a fluorescent cholesterol analog because it requires cholesterol mol% to be above a threshold of membrane enrichment to fluoresce. This creates a time lag that is readily apparent when comparing data in panels 2E/G and 2A. This could be addressed using more faithful analogs such a cholesterol alkynes. This is important to reconcile

because reliance solely on the D4H probe obscures whether the LDL cholesterol is trafficking directly to the recycling endosomes or being re-internalized from the PM. There is certainly ample PI(4,5)P2 at the PM for ORP2 to pick up in exchange for cholesterol.

We appreciate the Reviewer's point regarding a potential sterol re-entry route from PM to recycling endosomes and the suggestion to consider alkyne cholesterol as yet another probe for the study. A fluorescent sterol analog is indeed more sensitive than the D4H probe used in most of the experiments in the current work. Therefore, we employed the fatty acid (linoleate) ester of BODIPY-cholesterol incorporated into LDL particles as an alternative to trace the distribution of LDL-derived sterol. We have characterized and gained experience in using this probe in our previous work (Heybrock *et al.*, 2019; Kanerva *et al.*, 2013). We observed a concentration of BODIPY-cholesterol deriving from LDL in recycling endosomes at time points when BODIPY-cholesterol PM distribution was not evident. This, together with the acute pharmacological and genetic perturbations affecting the distribution of sterol between late and recycling compartments, proposes that LDL-derived sterol deriving from late endosomes rapidly reaches recycling endosomal compartments. However, considering that BODIPY-cholesterol (and likely other fluorescent sterol analogs) arriving at the PM will rapidly diffuse in the membrane, we would not readily detect small amounts of sterol that might arrive at the PM in parallel or even prior to we observe its more concentrated distribution in recycling endosomes. Therefore, we wish to leave open the possibility that ORP2 might act in lipid exchange between endosomes and the plasma membrane and have mentioned this in the discussion (p. 17-18).

Regarding the use of alkyne cholesterol as an alternative probe, it is an interesting candidate, but so far not well characterized for detecting dynamic changes in cellular sterol distribution, let alone LDL-derived sterol distribution. Moreover, after the click reaction and attachment of a fluorescent moiety, it becomes much less faithful a sterol analog once visible. To our knowledge, the first promising study where alkyne cholesterol ester was incorporated into LDL particles and successfully visualized in late endosomal compartments was recently reported by Dan Ory and Jean Schaffer (Feltes *et al.*, 2019). While this is indeed encouraging, alkyne cholesterol removal from endosomal compartments has not yet been demonstrated and in the published study, LDL uptake into cells was enhanced by overexpression of Scavenger receptor A (SR-A) and use of modified LDL particles. With this in mind, we piloted on labeling native LDL particles (used in the present study) with alkyne cholesterol and visualizing it upon clicking to a fluorophore. While some fluorescent labeling in endomembranes was detectable, the sensitivity was orders of magnitude lower than with BODIPY-cholesterol, and certainly not useful at this stage for solving the issue of whether the fluorescent sterol first arrives in recycling endosomes or at the PM. Thus, more work beyond the revision of the current study will be needed for setting up alkyne cholesterol for LDL-derived sterol trafficking studies.

3. The study is highly focused on ORP2 cholesterol trafficking from LE/Ly to recycling endosomes and does not provide a sufficient context for the broader audience to understand the relative contribution of ORP2 to post-lysosomal cholesterol movement. Does ORP2 KD affect delivery of cholesterol to organelles other than recycling endosomes? For example, what is the effect of ORP2 LOF in delivery of cholesterol to the ER for re-esterification or for suppression of SREBP2 target gene expression, or in delivery of cholesterol to the PM?

Thank you for raising these important points. As requested, we have now analyzed whether acute ORP2 degradation affects cholesterol delivery to the ER for re-esterification or suppression of SREBP2-target genes. These data reveal that acute ORP2 removal does not change the LDL-induced suppression of SREBP2 target gene (LDLR and HMGCR) expression (Fig 4H, I) or LDL-induced cholesterol esterification (Fig 4J) at early time points (0-4 h LDL labeling and ORP2 removal), when the effects of acute ORP2 loss on endomembrane cholesterol distribution (Fig 5) and LDL-cholesterol delivery to the PM (Fig 4F, G) are evident. These results suggest that the primary effects

of ORP2 on LDL-cholesterol trafficking are at the level of endosomes and routing towards the PM rather than towards the ER.

4. General concern: It is unclear from reading methods and figure legends whether specific IF panels are epifluorescence imaging or confocal slices. This is important in interpreting IF as to whether there is true overlap rather than superimposition.

Thank you for the careful scrutinization of our manuscript. This point was also brought up by Reviewer #2 and has now been addressed by providing additional details in Methods and Figure legends.

References

- Berginski ME, Vitriol EA, Hahn KM, Gomez SM (2011) High-Resolution Quantification of Focal Adhesion Spatiotemporal Dynamics in Living Cells. *PLoS ONE* 6: e22025
- Feltes M, Moores S, Gale SE, Krishnan K, Mydock-Mcgrane L, Covey DF, Ory DS, Schaffer JE (2019) Synthesis and characterization of diazirine alkyne probes for the study of intracellular cholesterol trafficking. *Journal of Lipid Research* 60: 707-716
- Hammond GR, Schiavo G, Irvine RF (2009) Immunocytochemical techniques reveal multiple, distinct cellular pools of PtdIns4P and PtdIns(4,5)P2. *Biochemical Journal* 422: 23-35
- Heybrock S, Kanerva K, Meng Y, Ing C, Liang A, Xiong Z-J, Weng X, Ah Kim Y, Collins R, Trimble W *et al* (2019) Lysosomal integral membrane protein-2 (LIMP-2/SCARB2) is involved in lysosomal cholesterol export. *Nature Communications* 10: 3521
- Kalappurakkal JM, Anilkumar AA, Patra C, Van Zanten TS, Sheetz MP, Mayor S (2019) Integrin Mechanochemical Signaling Generates Plasma Membrane Nanodomains that Promote Cell Spreading. *Cell* 177: 1738-1756.e1723
- Kanerva K, Uronen R-L, Blom T, Li S, Bittman R, Lappalainen P, Peränen J, Raposo G, Ikonen E (2013) LDL Cholesterol Recycles to the Plasma Membrane via a Rab8a-Myosin5b-Actin-Dependent Membrane Transport Route. *Developmental Cell* 27: 249-262
- Koponen A, Pan G, Kivelä AM, Ralko A, Taskinen J, Arora A, Kosonen R, Kari OK, Ndika J, Ikonen E *et al* (2020) ORP2, a cholesterol transfer, regulates angiogenic signaling in endothelial cells. *FASEB Journal* 158: 90-101
- Li S, Prasanna X, Salo VT, Vattulainen I, Ikonen E (2019) An efficient auxin-inducible degron system with low basal degradation in human cells. *Nature Methods* 16: 866-869
- Nader GPF, Ezratty EJ, Gundersen GG (2016) FAK, talin and PIPK1 γ regulate endocytosed integrin activation to polarize focal adhesion assembly. *Nature Cell Biology* 18: 491-503
- Panagiotakopoulou M, Lendenmann T, Pramotton FM, Giampietro C, Stefopoulos G, Poulikakos D, Ferrari A (2018) Cell cycle-dependent force transmission in cancer cells. *Molecular Biology of the Cell* 29: 2528-2539
- Salo VT, Li S, Vihinen H, Hölttä-Vuori M, Szkalicity A, Horvath P, Belevich I, Peränen J, Thiele C, Somerharju P *et al* (2019) Seipin Facilitates Triglyceride Flow to Lipid Droplet and Counteracts Droplet Ripening via Endoplasmic Reticulum Contact. *Developmental Cell* 50: 478-493.e479
- Santinho A, Salo VT, Chorlay A, Li S, Zhou X, Omrane M, Ikonen E, Thiam AR (2020) Membrane Curvature Catalyzes Lipid Droplet Assembly. *Current Biology* 30: 2481-2494.e2486
- Slack-Davis JK, Martin KH, Tilghman RW, Iwanicki M, Ung EJ, Autry C, Luzzio MJ, Cooper B, Kath JC, Roberts WG *et al* (2007) Cellular Characterization of a Novel Focal Adhesion Kinase Inhibitor. *Journal of Biological Chemistry* 282: 14845-14852
- Wang H, Ma Q, Qi Y, Dong J, Du X, Rae J, Wang J, Wu W-F, Brown AJ, Parton RG *et al* (2019) ORP2 Delivers Cholesterol to the Plasma Membrane in Exchange for Phosphatidylinositol 4, 5-Bisphosphate (PI(4,5)P2). *Molecular Cell* 73: 458-473.e457

EMBOJ-2020-106871

Takahashi et al. ORP2 couples LDL-cholesterol transport to FAK activation by endosomal cholesterol/PI(4,5)P₂ exchange

Thank you very much for taking the time and effort to help us satisfactorily address the remaining issues raised by Referee #2.

We submitted this manuscript to The EMBO Journal because of our experience of the Journal's responsible and highly professional editorial process. This is critically important now during the pandemic that carrying out experiments is much more difficult than normally. The first author of our study recently moved back to Japan after a 3-year postdoctoral stay in my group. This study is the main achievement of his postdoctoral work. For the manuscript revision, we managed to fly him back from Tokyo to Helsinki during the ongoing COVID-19 crisis. He stayed for a month (not counting the two-week quarantines) to carry out the experiments requested, with additional people in the lab helping to finalize them. We were very happy that in this way we managed to conduct all the key experiments proposed by the three Reviewers despite considerable restrictions on lab work. Furthermore, we were extremely satisfied that the additional data acquired strongly supported our original claims.

It seems to us that Reviewer #2 has a relatively focused biochemical mindset, is unlikely to change his/her mind and will continue to be unreasonably critical, which is now threatening to jeopardize this manuscript. Please note that we performed all the experiments originally requested by this Reviewer (see responses to the Reviewer's original queries in the attached file). We also wish to point out that the remaining points of Reviewer #2 deal with issues that the other referees had no problems with – even though these deal with general issues related to the specificity of chemicals and antibodies used.

I am here providing responses to the remaining issues and suggestions on how to respond to these to further improve the scientific quality of our manuscript.

Point-by-point responses to the remaining criticisms of Reviewer #2

1. As stated in the first review, experiments are using a relatively high level (10 micromolar) of a pharmacological inhibitor to FAK. Despite the authors providing references by others using this inhibitor concentration, this does not make it "correct". The PF-228 is not specific at the concentration used.

Response:

This may be the case, but to tackle this, we have included evidence in the revised manuscript that the effects of PF228 on LDL-derived cholesterol accessibility both in endomembranes and in the plasma membrane can be reproduced by FAK silencing. We can further strengthen these data (see response to Point 7). We referred to additional articles (Nader et al. Nat Cell Biol 2016, Panagiotakopoulou et al. Mol Biol Cell 2018, Kalappurakkal et al. Cell 2019) only to point out that the 10 micromolar concentration of PF-228 added to cells is commonly used and accepted in the field by authors, reviewers and editors of high impact journals.

We also wish to point out that PF228 has an IC₅₀ of 4 nM against the purified FAK protein (Slack-Davis et al. J Biol Chem 2007). It is normal that the inhibitor activity in cells is significantly lower (reflecting a reduced inhibitor concentration inside the cell or at a specific subcellular compartment) and cellular inhibitor concentrations in the range of 1-10 microM are not unusual even for highly potent inhibitors. PF228 is also a relatively specific FAK inhibitor: In Slack-Davis et al, 2007, it is demonstrated that PF228 has a >250 fold selectivity for FAK over the closest related kinase Pyk2 and 50 to >250 fold selectivity against all 41 tested kinases. Since we observe that ~10 microM inhibitor is required for efficient FAK inhibition in cells, inhibition of other kinases should be minimal at this concentration.

2. PF-228 is cell-permeable and will maximally inhibit FAK activity within 30 and maximally by 60 minutes (usually faster). A dose response curve (performed by western blotting is needed). In the materials provided to the reviewer, a graph is shown regarding pY397FAK intensity over time with inhibition at 50% at 4h. This is not standard. The authors need to show pY397FAK levels compared to total FAK (not a random loading control).

Response:

We have actually already done what the Reviewer asks for and provided a time and dose response curve of PF228 treatment by Western blotting of total cell lysates using pY397FAK for the reviewer in our response to his/her previous point 4 (see graph below). We apologize that we obviously did not specify this clearly enough in our response.

We used time points that are relevant in the setting of LDL-cholesterol delivery (several hours). This graph does make the point that the inhibitor needs to be used at a relatively high concentration also in the cells under study (although at 10 micromolar PF228, inhibition at 4h is about 70%, not 50%). We plotted pY397FAK immunoreactivity against protein loading in the same lane, because – as explained in response to the Reviewer's previous point 12 – stripping and reprobing of filters results in compromised specific recognition with these antibodies. Hence, total FAK intensity would need to be quantified from a separate blot and this would lower the accuracy of the result. We feel that adding these data to the manuscript would not be warranted as we are employing a commonly used concentration of the inhibitor (a response curve might be more relevant to include had we found that the compound is effective at a much lower concentration than e.g. used by Nader et al. in Nat Cell Biol 2016 that was one of the starting points of our study). However, if you see it necessary to add this image to the manuscript, it can be done.

3. The blots for the pY397 FAK antibody need to show non-cropped lanes. Non-specific reactivity to proteins others than FAK can occur with this antibody (as shown by authors). This raises key questions about what "real" signal is being detected within the high background of punctate spots.

Response:

We are submitting the whole immunoblots in the source data, as requested by EMBO Journal. In the last sentence of this comment, it seems that the reviewer is referring to immunofluorescence staining using anti-pY397FAK antibody, because he mentions high background of punctate spots. We respond to this in the following point (4) that continues on this topic.

4. The Appendix Figure S1 - is concerning because it shows that the speckled pattern of intracellular pY397FAK staining is non-specific after FAK siRNA knockdown. Loss of signal is at the cell peripheral spots that are likely focal adhesion sites. Where is the control for total FAK by cell staining?

Response:

There must be some misunderstanding here. The punctate staining pattern obtained with pY397FAK antibody is obviously and significantly reduced upon FAK siRNA knockdown. Minor background signal is typically observed by immunofluorescence staining, even when the specific epitope is missing from cells. This is inherent to the technique, with cellular autofluorescence, minor unspecific reactivity of

secondary antibody etc. contributing. Nevertheless, the strong immunoreactive signals both perinuclearly and at adhesion sites are markedly reduced in the knockdown cells. We did not include a control for total FAK staining, because this was not questioned by the Reviewer and we do now show cell stainings with this antibody.

5. Similar concerns with the b1 integrin siRNA - this 50% loss does not match the phenotypic effects. There is too much reliance on immunofluorescence. Both FAK and b1 integrin can be detected by blotting.

Response:

It is not clear what the Reviewer means with the phenotypic effects here, as we are not studying the phenotypic effects of beta1 integrin siRNA in this work. We have provided antibody controls for those antibodies that the Reviewer questioned and those applications that the antibodies were used for in our study. In his/her previous point 8, the Reviewer requested: "Need proof of specificity of anti-integrin and anti-pY397 FAK staining." We used anti-integrin antibody for fluorescence microscopy and anti-pY397FAK antibody for both fluorescence microscopy and Western blotting. We show in Appendix Fig. 1 that the anti-pY397FAK Western blotting signal is clearly reduced upon FAK silencing, evidencing for specificity. We further show in the same figure the specificity of the pFAK and beta1 integrin immunofluorescence stainings by demonstrating that siRNA knock-down results in a marked reduction in the fluorescence signals.

We did not use beta1 integrin antibodies for Western blotting. We did use FAK antibody for blotting. Although not originally requested by the Reviewer, we can add a control showing the specificity of the FAK antibody for Western blotting.

6. Since the authors have GFP-FAK-WT and GFP-FAK F397 constructs, they need to provide proof of signaling ORP2 signaling complex formation with intrinsic fluorescence or antibodies to GFP.

Response:

We do not understand where this entirely new request stems from. We have not claimed that ORP2 forms a signaling complex with FAK.

7. Figures 3C and EV3 panel B are from same cells and could be shown on the same panel. Could authors show the same panel with siRNA experiment? For some reason only Lamp1 and intracellular D4H staining is shown with control and FAK siRNA.

Response:

Yes, the Reviewer is correct that Figures 3C and EV3 panel B are from the same cells. We included EV3 panel B as an expanded view figure, because Figure 3 is already quite crowded. But we can show EV3B in the same panel if we move Figure 3A (PF228 effect on plasma membrane D4H intensity) into EV panels. We assume that by asking us to show the same panel with siRNA experiment, the Reviewer requests us to show that the D4H area positive for integrin beta1 is smaller in FAK siRNA than control siRNA treated cells. We should be able to add these data to Figure 3/EV Figure 3.

Summary of additional experiments proposed

- **Add antibody control for FAK immunoblotting (in Appendix Fig 1)**
- **Add data showing that FAK silencing recapitulates the reduced D4H (accessible cholesterol) signal in integrin beta1 containing organelles, analogously to PF228 in Fig 3F (new data to be generated and added either to Fig 3 or EV Fig 3)**

We are looking forward to your view regarding whether this constitutes a satisfactory revision plan for publication and are open to any suggestions you may have.

Thank you for submitting your revised manuscript to The EMBO Journal. I have read your response to our pre-consultation letter and also considered the points that we touched upon during our phone conversation.

Your action plan appears reasonable to me and, given the overall interest of your study, we have exceptionally decided to offer you the opportunity to revise the manuscript as indicated by reviewer #2 in a second round of revision. As stated in my earlier letter and reiterated this morning by phone, we feel that these remaining issues are important and have to be satisfactorily addressed. Depending on the quality and conclusiveness of the new results, we may either send the manuscript back to referee #2 or make an editorial decision.

Below, a summary of our discussion:

- Point 1: We agree on your proposal to test the specificity of PF228 inhibitor on LDL-derived cholesterol accessibility by silencing FAK. I also suggest you tone down the following statement in the rebuttal letter: "Since we observe that ~10 microM inhibitor is required for efficient FAK inhibition in cells, inhibition of other kinases should be minimal at this concentration". This is a speculation that has not been tested experimentally.
- Point 2: We agree that a PF228 dose-response curve for FAK phosphorylation (pFAK) should be performed in which the pFAK signal is normalized against the total FAK signal. You can detect pFAK and total FAK by loading the same sample twice and using different gels/blots, each having appropriate loading controls. Ideally, you could test a high and a low PF228 dose. If this is difficult because of the pandemic, please test at least the higher dose.
- Point 3: We agree that it would suffice to show the full anti-pY397FAK blot in the source data.
- Point 4: We agree with you including total FAK IFs as controls for FAK siRNA knockdown experiments.
- Point 5: We are satisfied with your plan to test beta-integrin and FAK knockdown by Western blot.
- Point 6: This point is not essential for publication and can be discussed in the rebuttal letter.
- Point 7: We find it acceptable to present Figure 3A and EV3B in the same panel, and that you add data showing that FAK silencing recapitulates the reduced D4H signal in integrin beta1 containing organelles.

Should you have any questions about the revision, please feel free to contact me. Also, I kindly ask you to promptly inform me in case a manuscript with a related content be published elsewhere.

I look forward to your revision.

Referee #1:

The authors have successfully addressed the points I raised, and I am happy to recommend publication of this revised manuscript in EMBO Journal.

Referee #2:

Revision of EMBO-J-2020-106871

The authors have improved their study with additional experimental results, details and clarification. However, big concerns remain about data collection and FAK.

1. As stated in the first review, experiments are using a relatively high level (10 micromolar) of a pharmacological inhibitor to FAK. Despite the authors providing references by others using this inhibitor concentration, this does not make it "correct". The PF-228 is not specific at the concentration used.
2. PF-228 is cell-permeable and will maximally inhibit FAK activity within 30 and maximally by 60 minutes (usually faster). A dose response curve (performed by western blotting is needed). In the materials provided to the reviewer, a graph is shown regarding pY397FAK intensity over time with inhibition at 50% at 4h. This is not standard. The authors need to show pY397FAK levels compared to total FAK (not a random loading control).
3. The blots for the pY397 FAK antibody need to show non-cropped lanes. Non-specific reactivity to proteins others than FAK can occur with this antibody (as shown by authors). This raises key questions about what "real" signal is being detected within the high background of punctate spots.
4. The Appendix Figure S1 - is concerning because it shows that the speckled pattern of intracellular pY397FAK staining is non-specific after FAK siRNA knockdown. Loss of signal is at the cell peripheral spots that are likely focal adhesion sites. Where is the control for total FAK by cell staining?
5. Similar concerns with the b1 integrin siRNA - this 50% loss does not match the phenotypic effects. There is too much reliance on immunofluorescence. Both FAK and b1 integrin can be detected by blotting.
6. Since the authors have GFP-FAK-WT and GFP-FAK F397 constructs, they need to provide proof of signaling ORP2 signaling complex formation with intrinsic fluorescence or antibodies to GFP.
7. Figures 3C and EV3 panel B are from same cells and could be shown on the same panel. Could authors show the same panel with siRNA experiment? For some reason only Lamp1 and intracellular D4H staining is shown with control and FAK siRNA.

Referee #3:

The authors have satisfactorily addressed concerns raised in the review.

EMBOJ-2020-106871R2

Takahashi et al. ORP2 couples LDL-cholesterol transport to FAK activation by endosomal cholesterol/PI(4,5)P₂ exchange

Point-by-point responses to the remaining concerns of reviewer #2.

1. As stated in the first review, experiments are using a relatively high level (10 micromolar) of a pharmacological inhibitor to FAK. Despite the authors providing references by others using this inhibitor concentration, this does not make it "correct". The PF-228 is not specific at the concentration used.

To tackle the potential unspecific effects of the FAK inhibitor PF-228, we have now included additional evidence in the revised manuscript that the effects of PF-228 on LDL-derived cholesterol accessibility both in the endomembranes and in the plasma membrane can be reproduced by FAK silencing (please see also response to point 7). We referred to additional articles only to point out that a 10 microM concentration of PF-228 added to cells is commonly used in the field. PF228 has an IC₅₀ of 4 nM against the purified FAK protein (Slack-Davis et al. J Biol Chem 2007). It is not uncommon that the activity of an inhibitor in cells is significantly lower (reflecting a reduced inhibitor concentration inside the cell or at a specific subcellular compartment) and cellular inhibitor concentrations in the range of 1-10 microM are not unusual even for highly potent inhibitors. PF228 is also a relatively specific FAK inhibitor: In Slack-Davis et al, 2007, it is demonstrated that PF228 has a >250 fold selectivity for FAK over the closest related kinase Pyk2 and 50 to >250 fold selectivity against all 41 tested kinases.

2. PF-228 is cell-permeable and will maximally inhibit FAK activity within 30 and maximally by 60 minutes (usually faster). A dose response curve (performed by western blotting is needed). In the materials provided to the reviewer, a graph is shown regarding pY397FAK intensity over time with inhibition at 50% at 4h. This is not standard. The authors need to show pY397FAK levels compared to total FAK (not a random loading control).

Thank you for the remark. Indeed, in the materials provided to the reviewer last time we plotted pY397FAK immunoreactivity against protein loading in the same lane, because stripping and re-probing of filters results in compromised specific recognition with these antibodies. Hence, total FAK intensity needs to be quantified from a separate blot. We have now done this and are providing a PF-228 dose response curve for FAK phosphorylation, where the pFAK signal is normalized against the total FAK signal, as requested, in the figure below. We have included the highest and lowest dose of PF-228 tested at time points that are relevant in the setting of LDL-cholesterol delivery (several hours). This graph makes the point that the inhibitor needs to be used at a relatively high concentration also in the cells under study.

3. The blots for the pY397 FAK antibody need to show non-cropped lanes. Non-specific reactivity to proteins others than FAK can occur with this antibody (as shown by authors). This raises key questions about what "real" signal is being detected within the high background of punctate spots.

This has been done; we are showing the full anti-pY397 FAK immunoblots in the source data, as requested by The EMBO Journal.

4. The Appendix Figure S1 - is concerning because it shows that the speckled pattern of intracellular pY397FAK staining is non-specific after FAK siRNA knockdown. Loss of signal is at the cell peripheral spots that are likely focal adhesion sites. Where is the control for total FAK by cell staining?

There seems to be some misunderstanding here. The punctate staining pattern obtained with pY397-FAK antibody is obviously and significantly reduced upon FAK siRNA knockdown. Minor background signal is typically observed by immunofluorescence staining, even when the specific epitope is missing from cells. This is inherent to the technique, with cellular autofluorescence, minor unspecific reactivity of secondary antibody etc. contributing. However, the strong immunoreactive signals both perinuclearly and at adhesion sites are markedly reduced in the knockdown cells. We have now included a control for total FAK by immunofluorescence staining using FAK siRNAs (Appendix Figure S1 C, representative images and D, quantification).

5. Similar concerns with the b1 integrin siRNA - this 50% loss does not match the phenotypic effects. There is too much reliance on immunofluorescence. Both FAK and b1 integrin can be detected by blotting.

We are not sure what the Reviewer means with the phenotypic effects here, as we are not studying the phenotypic effects of beta1 integrin siRNA in this work. We have now added FAK and beta1 integrin antibody controls by Western blotting using the relevant siRNAs (Appendix Figure S1 B).

6. Since the authors have GFP-FAK-WT and GFP-FAK F397 constructs, they need to provide proof of signaling ORP2 signaling complex formation with intrinsic fluorescence or antibodies to GFP.

We do not understand where this entirely new and surprising request stems from. We have not claimed that ORP2 forms a signaling complex with FAK in this manuscript.

7. Figures 3C and EV3 panel B are from same cells and could be shown on the same panel. Could authors show the same panel with siRNA experiment? For some reason only Lamp1 and intracellular D4H staining is shown with control and FAK siRNA.

The Reviewer is correct in that Figures 3C and EV3 panel B were from the same cells. We have now reorganized the figures so that they are shown in the same panel, as requested (new Figure 3A). Because this Figure is quite crowded, we moved the previous Figure 3A panel (PF-228 effect on plasma membrane D4H intensity) into EV Figure 3B, C. We have also added new data to show that FAK silencing recapitulates the reduced D4H signal in integrin beta1 positive organelles, as requested. These results are provided in the new Figure 3G (quantification) and EV Figure 3E (representative images).

Thank you for submitting your revised study. The manuscript has now been sent back to referee #2, whose comments are appended below.

As you will see, the referee finds that his/her criticisms remain not adequately addressed. However, the latest requests appeared excessive to us at this stage and we thus decided to contact an independent expert in the focal adhesion field, who had access to the manuscript, the referee's reports and the point-by-point letter.

Our expert finds that you have conclusively addressed the remaining points by referee #2 and recommends the manuscript for publication.

Please find below a list of editorial issues concerning the text and the figures that I need you to address before we can officially accept your manuscript.

Referee #2:

Revision of EMBOJ-2020-106871

Takahashi et al. ORP2 couples LDL-cholesterol transport to FAK activation by endosomal cholesterol/PI(4,5)P2 exchange

To the editor:

The authors have not addressed concerns that were raised in the initial and secondary review process. This reviewer has questioned the "over" interpretation of indirect immunofluorescence data presented by the author(s) with regard to FAK association with intracellular vesicles and FAK activation using antibodies that are shown by the authors to be "dirty" by western blot [new

supplemental data provided]. The fact that other bands [not FAK] react to the FAK and pYFAK397 antibodies (sometimes greater than the FAK band region) raise significant concerns for data interpretation. The pattern of high intensity FAK staining at immunofluorescent speckles in A431 cells is NOT something commonly observed in other cell types. A number of labs have published the use of FAK-null cells expressing various FAK mutants. These are freely available. The authors did not make use of this resource. The authors attempts to use FAK mutant over-expression were incomplete and missing validation of expression. The authors make strong statements in the Discussion such as "PI(4,5)P2 generation enhanced by FAK is probably one, if not the sole function of FAK in LDL-cholesterol delivery". This and others need to be revised as supporting data was not provided.

It is recommended that another independent review by a researcher in the FAK field be performed.

Specific comments to authors:

A number of previous specific comments were not necessarily addressed.

Point 1. Re: use of 10 micromolar FAK inhibitor treatment of cells.

This is a high concentration that will introduce non-specific effects. As the authors show that 0.5 micromolar treatment of cells results in the equivalent level of FAK inhibition after 4 hours, and many of the experimental cell measurement were made at a 4 hour treatment time point, the authors should show that 0.5 micromolar PF-228 treatment results is/are equivalent to 10 micromolar PF-228 treatment with respect to measured differences in lipid particle partitioning assays (or a dose response).

Point 2. See above.

Point 3. The addition of source data of the uncropped immunoblots for total FAK and pY397 FAK show random patterns of high background and non-specific band staining of multiple sizes. Sometimes the different sized bands are darker than the cropped FAK region bands. This remains concerning for effective interpretation of indirect immunofluorescence data.

Point 4. Figure 1 F, the pY397 FAK levels should be compared to total FAK. The total FAK blot is also missing from Appendix Figure S1 where equal FAK overexpression should be confirmed with both WT and kinase mutant Y397F construct.

Point 5. The authors show that b1 integrin siRNA knockdown has an inhibitory effect on their cell assay (phenotypic response). No controls were performed for potential knockdown effects of other integrins, as a specificity control for b1 integrin knockdown effects.

Point 6. FAK siRNA experiments need to be controlled by showing rescue of function by FAK wildtype (siRNA resistant) expression. This also needs to include a lack of kinase activity (FAK K454R) and a FAK autophosphorylation site mutant (FAK Y397F). This was requested as part of providing needed mechanistic support for the authors primary conclusions based on the use of a pharmacological inhibitor. The authors did not respond appropriately to this original request.

Moreover, as the authors make strong statements such as "PI(4,5)P2 generation enhanced by FAK is probably one, if not the sole function of FAK in LDL-cholesterol delivery" as in the Discussion, FAK

knockout and reconstituted cells have been characterized and are shared resources. The use of such cells would provide the still needed support for functional conclusions about the role of FAK activity.

Point 7. Addressed.

Suggestion: The schematic on Figure 8 could include the endosome names and although well explained in legend, the ORP2 role as transporter could be emphasized more. The same confusing impression of ORP2 role is presented in abstract: "Together, these results provide evidence that ORP2 controls FAK activation and LDL-cholesterol plasma membrane delivery by promoting bidirectional cholesterol/PI(4,5)P2 exchange between late and recycling endosomes." The phrase "... ORP2 controls FAK activation..." is too assumptive based on the authors response to point nr.6 saying that there is no proof of formation of signaling complex.

3rd Revision - Editorial Decision

3rd May 2021

I am pleased to inform you that your manuscript has been accepted for publication in The EMBO Journal.

Corresponding Author Name: Elina Ikonen

Manuscript Number: EMBOJ-2020-106871